# Stability of high-temperature salty ice suggests electrolyte permeability in water-rich exoplanet icy mantles

Jean-Alexis Hernandez [1,2,3]✉, Razvan Caracas [2,3,4] & Stéphane Labrosse [2]

Electrolytes play an important role in the internal structure and dynamics of water-rich satellites and potentially water-rich exoplanets. However, in planets, the presence of a large high-pressure ice mantle is thought to hinder the exchange and transport of electrolytes between various liquid and solid deep layers. Here we show, using first-principles simulations, that up to 2.5 wt% NaCl can be dissolved in dense water ice at interior conditions of water-rich super-Earths and mini-Neptunes. The salt impurities enhance the diffusion of H atoms, extending the stability field of recently discovered superionic ice, and push towards higher pressures the transition to the stiffer ice X phase. Scaling laws for thermo-compositional convection show that salts entering the high pressure ice layer can be readily transported across. These findings suggest that the high-pressure ice mantle of water-rich exoplanets is permeable to the convective transport of electrolytes between the inner rocky core and the outer liquid layer.

[1] European Synchrotron Radiation Facility, Grenoble, France. [2] CNRS, Ecole Normale Supérieure de Lyon, Université de Lyon, Laboratoire de Géologie de Lyon LGLTPE UMR 5276, Lyon 69364, France. [3] Centre for Earth Evolution and Dynamics, University of Oslo, Oslo 0315, Norway. [4] Université de Paris, Institut de Physique du Globe de Paris, CNRS, 1 rue Jussieu, Paris 75005, France. ✉email: jean-alexis.hernandez@esrf.fr

Water-rich exoplanets and icy moons are promising environments for the development of biological reactions. The extent and the composition of a potential ocean depends on the solubility and the availability of the various electrolytes. Both in situ measurements by planetary missions and aqueous alteration processes observed in chondrites suggest that salts based on $Na^+$, $Mg^{2+}$, $Cl^-$ and $SO_4^{2-}$ are present in extraterrestrial planetary aqueous environments[1–6]. While an interface between liquid water and a silicate mantle favors hydrothermal reactions releasing nutrients in the ocean, the presence of an underlying high-pressure ice mantle might prevent the transport of electrolytes[7] due to their insolubility in ice VI[8]. Nevertheless molecular ice VII has been shown to include non-negligible amounts of halides within their structures at conditions similar to large icy moon interiors[8–14]. These experimental results have recently been confirmed by the discovery of salty ice VII inclusions trapped in diamonds formed in the Earth mantle[15].

The ability of ice VII to retain salts in its structure results from its body-centered cubic (bcc) sub-lattice of O atoms, which is a common characteristic of all ice phases between 2 GPa and 300 K to at least 270 GPa and 2000 K[16]. At these conditions, bcc $H_2O$ ice undergoes multiple transitions due to changes in the distribution and the dynamics of the H atoms. Below 30–40 GPa and 800 K, ice VII (space group $Pn\text{-}3m$) is the stable phase. It has a static orientational disorder of the $H_2O$ molecules. The H atoms occupy the half-diagonals of the bcc O sub-lattice and the arrangement of $H_2O$ molecules form a strong hydrogen-bond (HB) network with O-H···O bonds, where O-H denotes the intramolecular covalent bond and H···O the intermolecular bond. Increasing pressure in ice VII results in a progressive dissociation of the $H_2O$ molecules due to the symmetrization of the O-H···O bonds. The compression first activates a dynamic translational disorder of H atoms along the O-H···O bonds (O-H···O ↔ O···H-O), this is ice VII′. Above 60–80 GPa, the symmetrization is completed, the H atoms are located at mid-distance between the two O atoms of the bcc half-diagonal, and the $H_2O$ molecules are fully dissociated; this ionic ice is labelled ice X[17,18]. Increasing the temperature in ice VII activates a dynamic orientational disorder of the $H_2O$ molecules[19]. Increasing both pressure and temperature leads to the combination of both orientational and translational motions of the H atoms and results in an onset of H diffusion characterizing the superionic regime[16,19–26].

These transitions strongly influence the compressibility, elasticity and transport properties of bcc ice[19,23,24,27–30]. For instance, the transition between superionic ice VII″ and ice VII′ is associated with a first-order transition, a dip in the elastic constants and a strong decrease of the electrical conductivity[19,24]. The completion the hydrogen bond symmetrization makes ice X stiffer than other bcc ices and preliminary investigation of its rheology predict a change of slip-system around 250 GPa[28].

The inclusion of salt in ice VII decreases its melting temperature[31] and breaks the connectivity of its hydrogen bond network[11,13]. In particular, the presence of halides increases the rotational disorder of water molecules and decreases the role of quantum effects in the O-H···O bond symmetrization[32], which shifts the transition to ice X by about 40 GPa at ambient temperature[12]. Nevertheless, the stability of salt-bearing ice is not known at conditions typical of water-rich super-Earths, i.e., up to a few megabars and few thousand Kelvin. The inclusion mechanism of NaCl in bcc ice remains debated and its influence on the development of the superionicity have not been investigated yet.

Here we compute the solubility and the properties of NaCl-bearing water ice along the 1600 K isotherm at relevant conditions for the interiors of water-rich exoplanets presenting a high-pressure ice mantle. Our approach is based on first-principles molecular dynamics simulations (DFT-MD) and extensive thermodynamic modeling. Then, based on our results and analytical considerations, we investigate how the presence of salt would influence the convection and the transport of electrolytes in the high-pressure ice layer of large water-rich exoplanets.

## Results and discussion

**Inclusion mechanism for $Na^+$ and $Cl^-$.** Several experimental studies[9,13,31] suggested three different and competing structural mechanisms for the inclusion of the $Na^+$ and $Cl^-$ ions in the bcc structure of ice VII. Based on experiments on NaCl-bearing ice, Frank[9] inferred that both $Na^+$ and $Cl^-$ ions occupy octahedral face-centered sites. Based on experiments on LiCl-bearing ice, another group[11,12,33] proposed that $Cl^-$ substitutes a water molecule and $Na^+$ remains in an octahedral site. Based on energy and size considerations, the same group suggested that both $Na^+$ and $Cl^-$ substitute $H_2O$ molecules on the bcc sites (double substitution mechanism). X-ray diffraction (XRD) measurements of natural (K,Na)Cl-bearing ice VII at ambient temperature[15] revealed the same substitution sites on the bcc lattice.

Consequently, as a prerequisite to our study, we determine the inclusion mechanism for $Na^+$ and $Cl^-$ ions in cubic water ice at 1600 K. We perform a direct crystallization of a salty water solution. We start with two equilibrated solutions in the fluid domain at 1600 K and 30 GPa, one NaCl·126$H_2O$ and one NaCl·30$H_2O$ (see Table 1 for notation and transformation between wt% vs molar%), and directly compress them at about 70 GPa, in the solid domain. The high-pressure simulations last for 12 picoseconds and both solutions crystallize into a bcc ice with both $Na^+$ and $Cl^-$ occupying the sites of the water molecules (Fig. 1). While the stoichiometry of the solutions may have favored the bcc arrangement, the outcome of these simulations could have been also very different, including a glassy state. But the fact that we eventually obtain bcc ice adds a further confirmation of the double substitution mechanism for incorporating salt impurities into dense ices, with both Na and Cl substituting $H_2O$ molecules.

**Properties and phase boundaries of the salty ice.** At each volume, we perform several DFT-MD in the canonical ensemble (volume and temperature fixed) with different configurations of Na and Cl in the bcc O sub-lattice and the resulting properties are averaged over the different configurations.

First, we evaluate the effect of NaCl inclusion on the bcc sub-lattice. The analysis of the radial pair distribution functions reveals that the substitution of water molecules by $Na^+$ and $Cl^-$ ions distorts the bcc sub-lattice (Fig. S13). At fixed volume ($a = 2.525$ Å), the shortest O-Na distance (i.e., the half-diagonal of the bcc unit-cell) is ~10% shorter than the O-O distance due to the stronger attraction between the two oppositely charged ions.

**Table 1 The equivalence between mole% and wt% for the various NaCl-bearing $H_2O$ terms used in the simulations denoted as NaCl·R$H_2O$.**

| Mole fraction NaCl | Mass fraction NaCl | R | Supercell | $N_{NaCl}$ |
|---|---|---|---|---|
| 0.004 | 0.013 | 248 | 5 × 5 × 5 | 1 |
| 0.008 | 0.025 | 126 | 4 × 4 × 4 | 1 |
| 0.016 | 0.050 | 62 | 4 × 4 × 4 | 2 |
| 0.019 | 0.059 | 52 | 3 × 3 × 3 | 1 |
| 0.032 | 0.098 | 30 | 4 × 4 × 4 | 4 |
| 0.067 | 0.189 | 14 | 4 × 4 × 4 | 8 |
| 0.149 | 0.351 | 6 | 4 × 4 × 4 | 16 |

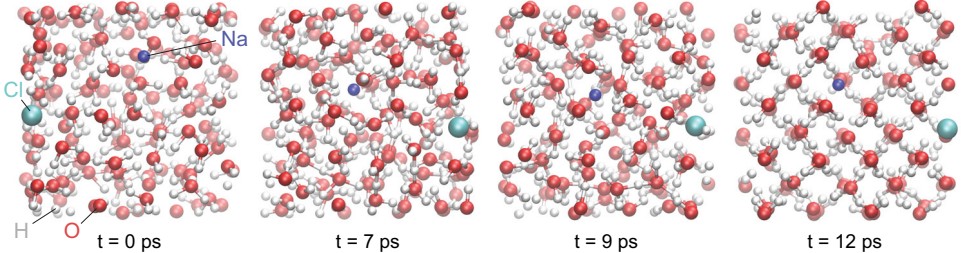

**Fig. 1 Crystallization of a NaCl·126H$_2$O solution during a DFT-MD simulation at 70 GPa.** The simulations are done in the NVT ensemble (constant number of particles, volume and temperature). During crystallization both Na$^+$ and Cl$^-$ ions occupy body-centered cubic lattice sites and both substitute a water molecule.

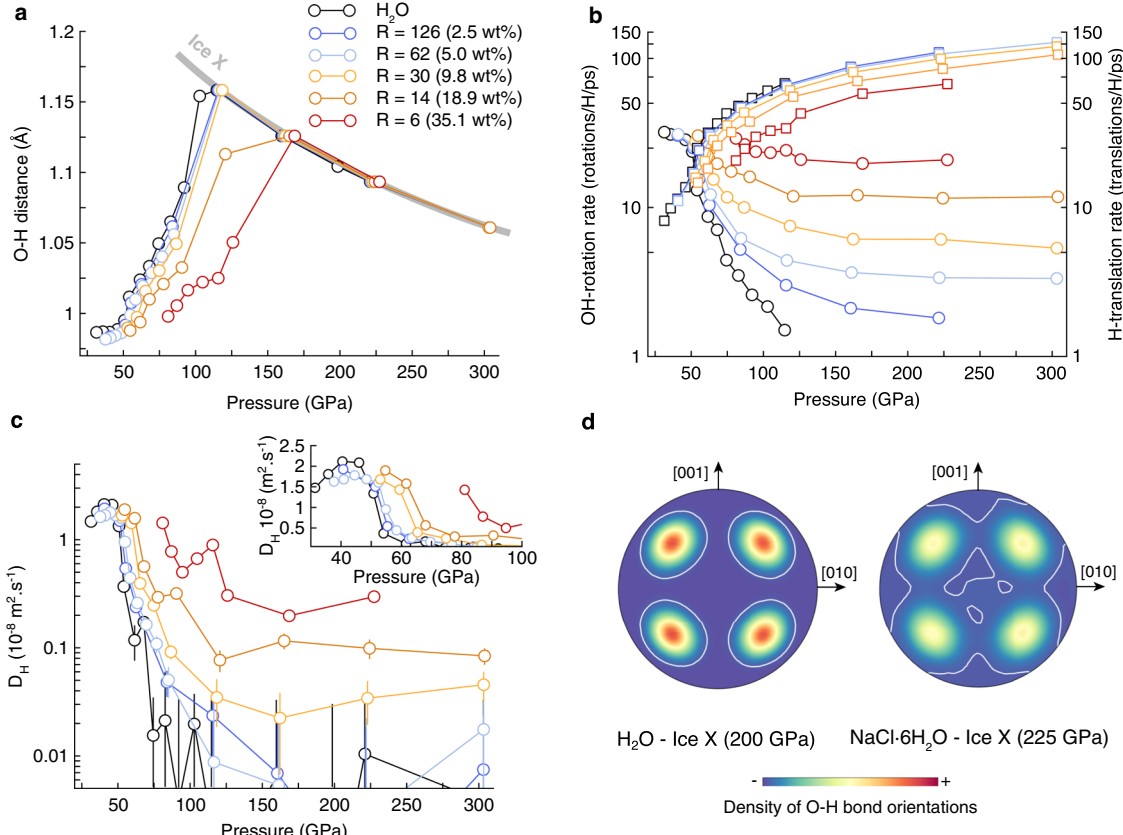

**Fig. 2 H-bonding regimes in NaCl-bearing bcc ice at 1600 K. a** Pressure dependence of the shortest O-H bond length. For a given NaCl concentration, OH reaches a maximum at the transition towards ice X. See text for more details. **b** OH-rotation rates (dots) and H-translation rates (squares) under compression in salty ice. **c** H diffusion coefficients ($D_H$). The inset represents a zoom in the 30–100 GPa pressure range showing the increasing pressure of the onset of diffusion with increasing NaCl content. When non-visible error bars are smaller than the size of the markers. **d** Stereographic projection of the density distribution of O-H bond orientations in ice X. In pure and salty, the four maxima corresponds to the ⟨111⟩ directions as expected for O-H bonds aligned with the half-diagonals of the bcc sub-lattice. In salty ice the orientation distribution of the O-H bonds is broader than in pure ice, indicating that H atoms tends to delocalize more out of the ⟨111⟩ direction (OO axis) than in pure ice. The color scale is the same for the two distributions and the white lines contours the same density of O-H bond orientations.

O and Cl, having charge with the same sign, repulse each other resulting in longer O-Cl bonds. At low densities (2.361g.cm$^{-3}$) and for a given NaCl concentration (NaCl·30H$_2$O), the O-Na and the O-Cl distances respectively differ from the O-O distance by $-14\%$ and by $+6\%$. The differences change to $-9.5\%$ and $+8\%$ at higher densities (4.221g.cm$^{-3}$). These distortions of the bcc sub-lattice are expected to decrease the elastic constants of NaCl-bearing ice compared to pure water ice.

As mentioned above, the various phases of bcc ice differ by the distribution and the dynamics of the protons (H atoms). To identify the different regimes, we perform a three-body analysis[19,24] of the O-H⋯O bonds based on the atomic trajectories.

The comparison of the shortest OH distance to the shortest OO distance allows us to monitor the symmetrization of the O-H⋯O bonds, and the transition to ice X when OH = OO/2. Figure 2a shows that increasing concentration of NaCl shifts the transition to ice X towards higher pressures compared to pure ice. For instance, there is an upwards shift of 50 GPa in the ice X transition pressure when adding 35.1 wt% NaCl.

Counting the number of H-translations ($O_a$-H⋯$O_b \rightarrow$ $O_a$⋯H-$O_b$) and OH-rotations ($O_a$-H⋯$O_b \rightarrow O_a$-H⋯$O_c$)

allows us to quantify dynamical disorders and to distinguish all other bcc phases in our simulations. Ice VII is not dynamically disordered and both OH-rotation and H-translation rates would be equal to zero. Plastic ice (free rotations of $H_2O$ molecules) would have a positive OH-rotation rate and a H-translation rate equal to zero. Non-diffusive ices VII′ and X would have H-translations without any OH-rotation. The presence of both H-translations and OH-rotations indicates that H atoms are diffusing: this characterizes the superionic regime. In this case, superionic ice VII″ is distinguished from superionic ices VII′ and X by a OH-rotation rate greater than the H-translation rate. Figure 2b shows that in pure $H_2O$ ice both H-translations and OH-rotations are activated up to ~150 GPa, indicating that H atoms diffuse below this pressure at the time-scale of our simulations and that pure ice is in a superionic regime. Below 52 GPa, pure ice is affected by more OH-rotations than H-translations and is in the superionic VII″ phase. Above 150 GPa, only H-translations are observed in pure ice indicating that ice is in its non-diffusive ice X form (OH = OO/2 at these conditions).

At a given pressure, increasing concentrations of NaCl strongly enhances the orientational disorder in bcc ice (increase of the OH-rotation rate and decrease of the H-translation rate). As a consequence, the pressure at which the OH-rotation rate equals the H-translation rate, i.e., the transition pressure between superionic ice VII″ and superionic ice VII′ (or superionic ice X depending on the relation between the OH and OO distances), increases with the NaCl concentration. Also, above 2.5 wt% NaCl, we found that the OH-rotation rate never reaches zero, indicating that salty ice is superionic at all sample pressure at 1600 K. The enhanced orientational disorder in salty ice is caused by both the inability of Na to form any bonds with H, and by the absence of covalent Cl-H bonds. Only O-H···Cl bonds exist and present a strong asymmetry compare to O-H···O bonds as both the O-H bond and is H···Cl bonds are ~20–25% shorter than in normal O-H···O bonds (see Fig. S14).

These changes in bonding regime are reflected in the variation of the proton diffusion (Fig. 2c) and electrical conductivity $\sigma_e$ (see Fig. S15). In particular, superionic ice VII″ is characterized by the highest diffusion coefficients with a plateau at 1.5–2 $m^2.s^{-1}$ ($\sigma_e$ =40–60 S.cm$^{-1}$) in both pure and salty ices. Under pressure, $D_H$ and $\sigma_e$ decrease down to statistically zero (i.e., $D_H < 10^{-10}$ $m^2.s^{-1}$) in ice with NaCl concentrations lower than 5.0 wt% although both OH-rotational and H-translational disorder still persist (Fig. 2b). For large NaCl concentrations ($R = 14$ and 6), $D_H$ presents a second maximum at 100–120 GPa related to the enhanced H delocalization out of the O − H···O bonds (Fig. 2d).

In summary, the inclusion of salt distorts the bcc sub-lattice, increases the orientational disorder and weakens the hydrogen-bond network of the ice. This shifts the superionic VII″ - VII′ - X transition sequence towards higher pressure with increasing salt concentrations (see Fig. 3). Therefore, at given pressure-temperature conditions, salty ice is less stiff, has a greater electrical conductivity and is likely to be less viscous than pure ice.

**Solubility of NaCl in high-temperature water ice.** The entire phase diagram of ice at high pressure and temperature is affected by the presence of salt, with the phase boundaries pushed to higher pressures (Fig. 3). The amplitude of the shift depends on the amount of salt that can actually be dissolved into the ice. Its determination needs to pass through the evaluation of the Gibbs free energies for the ice-salt solid solutions. This involves the estimation of the mixing terms, which can considerably lower the energy of the intermediate terms with respect to the mechanical mixture of the pure end-member terms.

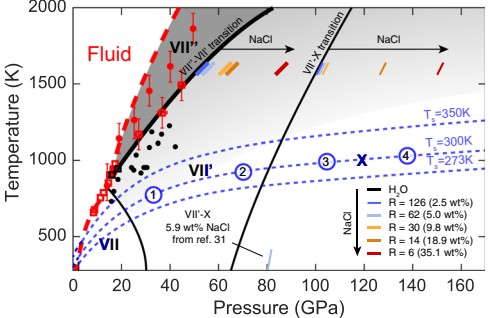

**Fig. 3 Phase diagram of bcc water ices.** Solid-solid phase boundaries in pure ice are indicated by black lines. Corresponding phase boundaries in NaCl-bearing ice (NaCl·$RH_2O$) ice are shown at 1600 K and 300 K. Grey areas indicate where pure $H_2O$ is superionic, with darker areas corresponds to regions with higher H-diffusion coefficients. The phase boundaries in pure $H_2O$ ice are drawn based on our DFT-MD simulations[19,24] and on both experimental[64] and theoretical works[18,65] that take into account quantum effects at 300 K. The melting line of pure ice as obtained in our single-phase simulations is shown by the red dashed line. Experimental melting points from Schwager et al.[66]. and Queyroux et al.[26]. are shown as red circles and red empty squares respectively. Black empty squares correspond to the experimental determination of first-order transition between superionic ice VII″ and ice VII′[26], and black dots refer to a solid-solid transition[66], likely corresponding to the same transition. Blue dashed lines show adiabatic profiles in the $H_2O$ layer of hypothetical ocean exoplanets with habitable pressure-temperature conditions in the surface ocean and no or thin atmosphere (see Supplementary Information). The different profiles are anchored at surface temperatures of 273 K (temperature of crystallization of pure ice Ih), 300 K and 350 K (representative temperature for an optimal growth of thermophile and hyperthermophile organisms[67]) respectively. Circled numbers 1, 2, 3 and 4 indicate approximate pressures at the bottom of the high-pressure ice mantle for 1, 2, 3, and 4 $M_\oplus$ planets (i.e., ~1.3, 1.5, 1.7, 1.9 $R_\oplus$) with 50 wt% $H_2O$ and Earth-like cores as calculated in Sotin et al. (2007) for a surface temperature of 300 K.

In solid solutions, the configurations that present the major contribution are generally separated by high energy barriers although their own energies can be very close or similar. As such, the transition paths between configurations cannot be sampled directly by standard molecular dynamics. Then statistical methods need to be employed to estimate the configuration entropy, as described in the methodology section, which depends on the number of configurations accessible to the system, i.e., the Na, Cl, and O arrangements in the bcc sub-lattice. As we investigate the solubility and properties of salty ice along the 1600 K isotherm, we validate the statistical approach by a second set of calculations exploring all possible arrangements for two concentrations at 100 GPa and 1600 K.

At 1600 K and 100 GPa, the Gibbs free energy of mixing is negative up to 5–10 wt% NaCl depending on the estimation of the configuration entropy and presents a minimum at 2.5 wt% NaCl (NaCl·$126H_2O$). Assuming that $H_2O$-bearing B2-type NaCl is the only phase that forms close to the NaCl end-member term, and that its free energy of mixing has only a small negative incursion, the common tangent to the two free energy curves intersects the curve of NaCl-bearing $H_2O$ free energy of mixing close to its minimum. This suggests that the inclusion of NaCl in water ice at 1600 K between 80 and 200 GPa is thermodynamically favorable up to ~2.5 wt% NaCl. At these conditions, salty superionic ice always presents both a larger volume (by +0.41% for 2.5 wt% NaCl and by +0.86% for 5.88 wt% NaCl, both at 1600 K and

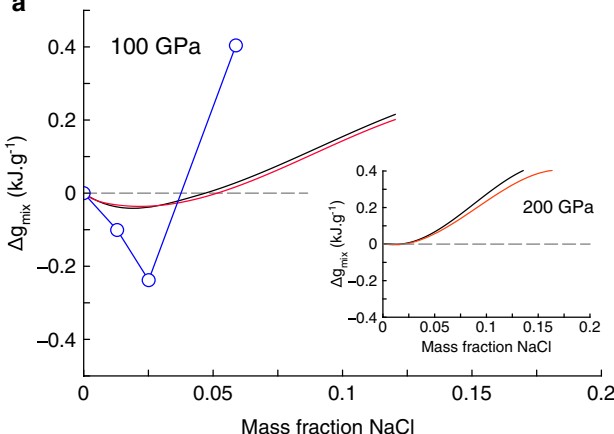

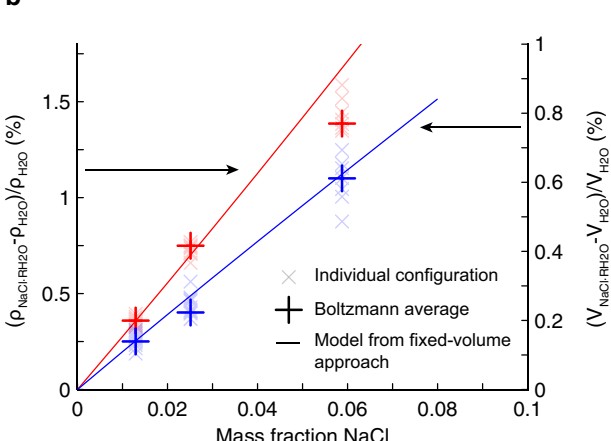

**Fig. 4 Thermodynamics of NaCl·$R$H$_2$O mixture at 1600 K and 100 GPa. a** Gibbs free energy of mixing $\Delta g_{mix}$ as computed from a fixed pressure approach and a complete sampling of the reduced configuration space (blue circles). In this case the configuration entropy is exact. Black (with correction for the GGA volume overestimation in B2 NaCl) and red (without) lines correspond to $\Delta g_{mix}$ calculated from the fixed volume approach (see text and Supplementary Information) and a random sampling of the entire configuration space. Due to the limited sampling, the configuration entropy is less constrained in this approach. Inset: $\Delta g_{mix}$ at 200 GPa and 1600 K. **b** Relative volume (blue) and density (red) differences between NaCl · $R$H$_2$O and H$_2$O. Crosses corresponds to individual configurations and pluses to Boltzmann averages obtained from the complete sampling of the reduced configuration space. Lines correspond to the model derived from the fixed volume approach.

100 GPa) and a larger density than pure water ice (Fig. 4a). Such amounts are in agreement with recent experimental and theoretical works performed on the inclusion of various halides (NaCl, LiCl, MgCl$_2$, RbI) in ice VII at lower pressures and temperatures[8,9,11–14]. A solubility of 2.5 wt% is still consistent with the observation of temperature-induced exsolution of some NaCl above 500 K in NaCl-doped ice VII whose initial concentration was higher[10]. The volume increase due to the addition of salt contrasts with the XRD measurements of Frank et al.[9,10]. but agrees with all previously mentioned studies[8,11–13].

**Transport of electrolytes in a high-pressure ice mantle.** Salts are observed at the surface of small icy bodies of the Solar system and may be produced from hydrous alteration processes taking place before and during the accretion, and/or from the interactions between the water-rich layer and a rocky core. Although our DFT

calculations are done at 1600 K, the applications of our results are not restricted to this exact temperature. Indeed, the combination of our high-pressure and high-temperature results with previous studies done at milder conditions[8,13] shows that bcc ices are able to include non-negligible amounts of salt in their structure over a broad range of conditions relevant for the interiors of water-rich super-Earths[34] ($1R_\oplus < R < 1.8R_\oplus$), mini-Neptunes ($1.8R_\oplus < R < 4R_\oplus$), and Neptune-like exoplanets[35,36], according to the classification made in ref. [37,38]. based on the Kepler exoplanet catalogue.

As shown above, the inclusion of salt affects the phase diagram and the transport properties of bcc ices (e.g., extension of the superionic field and of the transition pressures between the different phases) and increases the density by about 0.5% compared to pure ice, as the positive change in specific volume due to the addition of salt is compensated by its higher molecular weight. These changes are expected to influence the dynamics of a high-pressure ice mantle in a variety of different ways, depending on the planet characteristics. A full understanding of salt transport between the rocky core and the ocean by convection in the high-pressure ice layer requires complex fluid dynamical double-diffusive models including complexities such as phase changes, pressure-, temperature- and composition-dependence of physical properties, and possibly partial melting[39–41]. This falls clearly beyond the scope of the present paper but, the feasibility of the process can be evaluated using the current knowledge on mantle convection, which includes similar ingredients[42].

Given that salts and other electrolytes are a necessary (but of course not sufficient) condition for habitability in a (sub-)surface ocean, it is interesting to evaluate if salt can be involved in a global cycle, i.e., if salt and other electrolytes could be transported throughout the high-pressure ice mantle by convection. In the following, we develop our reasoning and provide numerical applications to a hypothetical planet of 1 M$_\oplus$ with 50 wt% H$_2$O, an Earth-like core, absent or very thin atmosphere, and a surface temperature of 300 K (case 1 in Fig. 3) allowing for the existence of a surface ocean with mild enough temperature conditions to sustain life as we know it[43]. Such case (~1.3 $R_\oplus$) is chosen to fit within the possible range of super-Earths ($R < 1.6 - 1.8R_\oplus$). It is depicted in Fig. 5, and presents a direct ocean/ice VII boundary as ice VI melts below 350 K. Planets with radii above the so-called radius valley ($R > 1.8R_\oplus$) are thought to have thick atmospheres which would result in higher temperatures in the ocean[37,38]. However, it is worth mentioning that the case of larger water-rich super-Earths with similar surface ocean conditions cannot be excluded. Exoplanets with orbital periods longer than 100 days and receiving less than 10 times the Earth's Solar flux are not observed because of biases in observations (see Supplementary Information for a discussion on statistical results based on the Kepler catalogue). It is depicted in Fig. 5, and presents a direct ocean/ice VII boundary as ice VI melts below 350 K.

Considering an adiabatic profile in the H$_2$O layers (see Fig. 3, and Supplementary Information for calculation details), we find that the temperature at the bottom of the ocean is 367 K, resulting in a liquid/ice VII boundary at the bottom of the ocean and about 800 K at the bottom of the ice mantle in the absence of thermal boundary layer with the underlying rocky mantle. This indicates that the whole high-pressure ice mantle is composed of bcc ice as depicted in Fig. 5. Thermal convection in a high pressure ice layer depends on the value of the dimensionless Rayleigh number which, in the case of an imposed heat flux $q_c$ from the rocky core, takes the form[39]:

$$Ra = \frac{\alpha \rho g q_c d^4}{k\kappa\eta} = 3 \times 10^{13} \left(\frac{R_c}{5000\,\text{km}}\right)\left(\frac{q_c}{10\,\text{mW/m}^2}\right)\left(\frac{d}{3000\,\text{km}}\right)^4\left(\frac{\eta}{1.3 \times 10^{17}\,\text{Pa s}}\right)^{-1}$$

$$(1)$$

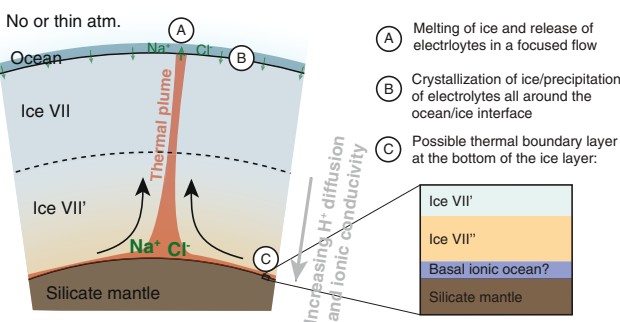

**Fig. 5 Transport of salt through the high-pressure ice mantle of a hypothetical water-rich exoplanet with 1 $M_\oplus$, 50 wt% $H_2O$, and a surface temperature of 300 K (corresponds to case 1 in Fig. 3).** A thermal plume creates a focused upward flux of salty ice that melts at the boundary with the ocean. The crystallization at the bottom of the ocean over a broad area produces a diffuse return flow of salt in the mantle. Depending on the initial conditions and on the partitioning coefficients of NaCl between the ice and the ocean, the icy mantle may acts either as a well or a source of electrolytes for the ocean. Label C illustrates a possible thermal boundary layer at the bottom of the icy mantle at the contact with the hotter rocky mantle.

with $\alpha$ the coefficient of thermal expansion, $\rho$ the density, $g$ the gravitational attraction, $d$ the thickness of the ice layer, $\eta$ the viscosity, $k$ and $\kappa$ the thermal conductivity and diffusivity, respectively. However, transport properties for bcc ices are quite uncertain[28]. To our knowledge no viscosity measurement in ice VII has been reported yet and we highly encourage any experimental effort in this direction. Here we use values quoted for ice VI from Choblet et al.[39] as a guideline and keep on the right-hand-side of Eq. 1 the parameters most uncertain or varying among planets. In the present case, geometric factors of the planet are from Sotin et al.[44], but these values can be changed depending on the planet of application. The gravity acceleration is computed assuming a rocky core of radius $R_c$ and density $\rho_c = 6 \times 10^3 \text{kg.m}^{-3}$. The reference values for the various parameters are quite conservative and the Rayleigh number is expected to be well above the critical value $Ra_c = 352$ for the onset of solid state convection in the presence of a solid-liquid phase change at the surface[45], validating the choice of an adiabatic temperature profile in the bulk of the ice layer. The lowermost part of the layer would see a steeper temperature gradient associated with the convective boundary layer. For such large values of $Ra$, the typical flow velocity in the ice layer scales as[46] $V_{rms} = 0.2 Ra^{0.66} \kappa/d \sim 700 \text{m/yr}$ for the reference values used above, which can very efficiently transport salty ice from the ice/core interface to the ocean. The most important shortcoming in these estimates is neglecting the effect of salts on density which should impede thermal convection if we consider that it enters at the bottom of the solid. This effect can be quantified by introducing the buoyancy number, $B = \Delta\rho_\chi / \rho\alpha\Delta T$, with $\Delta T$ the temperature difference driving thermal convection and $\Delta\rho_\chi/\rho$ the density increase owing to salt enrichment[47]. A large value of $B$ leads to a strong effect of composition and possible stratification, while efficient entrainment is possible for $B \ll 1$. The exact value for the transition in regimes has not been studied for the problem discussed here but is generally around 1. For convection in the solid ice layer surrounded by an ocean, $\Delta T$ is the temperature drop across the bottom boundary layer[46] and can be estimated from the scaling of heat transfer as function of the Rayleigh number (see Methods). The value of $\Delta\rho_\chi/\rho$ must be lower than

the solubility limit, $\delta\rho_m = 0.7\%$, but also depends on the mode of transfer from the rocky core. The largest flow of salt is likely to happen if ice melting occurs at the interface with the rocky core. In this case, salts may be transported to the ocean by porous flow[39–41]. We here consider the situation, a priori less favorable for transfer of salts to the ocean, where no melting occurs and salts flow to the ice layer by diffusion, with a diffusion coefficient $D$. The relevant value entering $B$ is the density increase from salts in the thermal boundary layer, with a thickness $\sqrt{\kappa t_\star}$ with $t_\star$ the time needed to make the boundary layer unstable[48]. In the same time, the concentration of salts increases in a layer of thickness $\sqrt{D t_\star}$ making the average relative density increase from salts in the thermal boundary layer equal to $\delta\rho_m \sqrt{D/\kappa}/2$. Combining all that (see Methods) leads to:

$$B = 3 \times 10^{-2} \left( \frac{D}{3.5 \times 10^{-12} \text{m}^2\text{s}^{-1}} \right)^{1/2} \left( \frac{R_c}{5000\text{km}} \right)^{1/3} \left( \frac{q_c}{10\text{mWm}^{-2}} \right)^{-2/3} \left( \frac{d}{3000\text{km}} \right)^{1/3} \left( \frac{\eta}{1.3 \times 10^{17}\text{Pas}} \right)^{-1/3} \quad (2)$$

Taking the same reference values as before and a tentative value for the maximum salt diffusivity equal to that of O atoms, $3.5 \times 10^{-12} \text{m}^2 \text{s}^{-1}$ (see Methods), leads to a very small buoyancy number which implies a very limited effect of salts against thermal convection. Moreover, the style of convection expected in an ice layer bounded above by an ocean shows focused hot plumes with a diffuse return flow[46]; the up-welling material melts when it reaches the ocean[39,45], therefore releasing salts there (Fig. 5). The presence of a high pressure ice mantle in a water planet should therefore not be considered as a chemical insulator for the transport of NaCl and likely other electrolytes, towards a shallow ocean[39]. We suggest future assessments of the habitability potential of large water-rich planets to consider the present results.

Both the efficiency and the duration of the electrolyte transport to the shallow ocean depend on the actual size, composition and surface temperature of the considered planet, which might result in different scenarios at the interface between the ocean and the ice mantle, and between the ice mantle and the rocky core[8,44,49,50]. In the case of an exoplanet with a thick atmosphere, most of the ocean would reach a supercritical state, and the habitability would not be preserved. In such planets, the ocean would also be in contact with bcc ice (VII ou superionic VII″ depending on the temperature) given that the total amount of $H_2O$ is large enough to reach the liquid/solid transition. If the surface temperature is very high, the conditions in the water layer may never intersect the melting line. In the case of an exoplanet with a very low surface temperature without atmosphere, a layer of ice VI could cap the ice VII layer, and the phase change VII to VI in up-welling plumes could lead to ex-solving NaCl to make a polycrystalline assemblage[8]. The gross density of the assemblage should be similar to that of salt-doped ice VII and, according to the discussion surrounding Eq. (2), should not prevent convection to proceed.

Finally, depending on the planetary formation scenario, the initial electrolyte concentration in the water-rich envelopes may vary. If water is brought after the crystallization of the rocky mantle, water may remain relatively free from alteration products. If the accretion of the icy and rocky materials is synchronous, the water-based layers would likely contain a certain initial concentration of electrolytes due to the production of salts by hydrous alteration of chondritic material[6]. In this case, the evolution of the ocean salinity would depend on the partitioning of salts between liquid water and ice during crystallization at the bottom of the ocean. The enrichment in salt of the icy mantle

would be influenced by the nature of the water/rock interface. In the case of a solid-solid interface, the enrichment of the ice layer depends on the partitioning coefficients of each chemical species between the silicates and the cubic ice. If partial melting occurs at the bottom of the high-pressure ice mantle due to the temperature increase in a thermal boundary layer, the incorporation of impurities in the ice could be much more efficient than in the case of a solid-solid interface[39–41]. Future studies need to estimate the partitioning coefficients of essential electrolytes between liquid water, superionic ice, and silicates at conditions comparable to the base of the ice layer in ocean planets to improve our understanding of the internal functioning of these planets and to better assess their habitability.

## Methods

### First-principles molecular dynamics

*Computational details.* We study the behavior of NaCl-bearing dense $H_2O$ ice by performing first-principles molecular dynamics (DFT-MD) simulations with the Vienna Ab initio Simulation Package[51,52] (VASP v4.6 and v5.2). To model the atoms, we use the PAW potentials[53,54] with the following electronic configurations: O:[He]$2s^2 2p^4$, H:$1s^2$, Na:[Ne]$3s^1$ and Na:[He]$2s^2 2p^6 3s^1$, and Cl:[He]$2s^2 2p^5$. The electronic structure is calculated within the density functional theory (DFT) framework and the Perdew-Burke-Ernzerhof general gradient approximation[55] is used to approximate the exchange-correlation term. Most DFT-MD simulations are performed with $4 \times 4 \times 4$ bcc supercells (see Table 1) with a cutoff energy of 550 eV for the plane-waves that ensures that internal energy is converged within 5 meV per atoms, and 1.5 GPa precision in pressure with respect to a calculation with a cutoff energy of 1000 eV. We calculate electronic wavefunctions at the Γ-point of the Monkhorst k-point grid[56]. Time steps of 0.5 or 1 fs are employed in all simulations. DFT-MD are either performed in the NVT, i.e., constant Number of particles, Volume, and Temperature, and NPT, i.e., constant Number of particles, Pressure, and Temperature, ensembles. For NVT sampling, temperature is controlled using velocity rescaling (isokinetic) as no difference was observed for averaged properties when using a Nose-Hoover thermostat[57,58]. Isobaric-isothermal simulations are performed with Raman-Parrinello molecular dynamics[59] and a Langevin thermostat.

*Determination of the free energy of mixing.* The estimation of the thermodynamic stability of the binary system requires the calculation of the Gibbs free energy of mixing $\Delta g_{mix}(w)$ as a function of salt concentration $w$, pressure $P$, and temperature $T$. $\Delta g_{mix}(w, P, T)$ is related to the Gibbs free energies of the pure phases ($g_{H_2O}(P, T), g_{NaCl}(P, T)$) and to the configuration averaged Gibbs free energy of the mixture ($g_{sol}(w, P, T)$) as:

$$\Delta g_{mix}(w, P, T) = g_{sol}(w, P, T) - (w g_{NaCl}(P, T) + (1 - w)g_{H_2O}(P, T)) \quad (3)$$

$g_{sol}(w, p, T)$ includes a configuration entropy term, $s_{conf}(w, P, T)$.

First we compute the free energies of the two end-members: $H_2O$ and B2-type NaCl. We estimate the vibrational contribution to the entropy and the corrections for the nuclear quantum effects by computing the partial vibrational spectra from the velocity autocorrelation functions and using the two phases thermodynamic memory function model (2PT-MF)[23,60–62]. We estimate the configuration entropy for the $H_2O$-rich part of the solid-solution taking advantage of recent developments in the computation of mixture thermodynamics based on reduced configuration space sampling[63].

Then we perform a series of NVT simulations with different concentrations of NaCl at different volumes (Table 1); at each given volume we consider several but not all possible starting NaCl configurations that yield an average Helmholtz free energy $F$ from which we deduced the configuration entropy and weighted thermodynamic properties. From the resulting data points, we construct the Helmholtz free energy surface at 1600 K over the entire volume range of interest. By derivation we obtained the Gibbs free energy of the solution as function of pressure and NaCl concentration (see section S2).

Nevertheless as we sample a subset of the configuration space, the configuration entropy is likely underestimated. Second we perform NVT simulations at 100 GPa for all possible configurations at three NaCl concentrations ($R = 250, 126$ and $52$) to have a correct value of the configuration entropy; the two approaches yield about 5% difference in the mass fraction of NaCl that can be dissolved in the ice.

The full description of the free-energy estimation and the fitting procedure to construct the free energy surface are available in Supplementary Information. A numerical implementation can be found at https://osf.io/w64fm/.

*Estimation of NaCl diffusivity.* The diffusivity of NaCl ($D$) in the ice is required to calculate the buoyancy number $B$ of a thermal boundary layer enriched in salt at the bottom of the ice mantle. In order to provide a reasonable order of magnitude for this quantity, we consider that $D$ is similar to the diffusivity of O atoms $D_O$, possible in our calculations only in the presence of vacancies. To estimate $D_O$, we

perform an additional DFT-MD run in a defective ice X structure with 126 $H_2O$ and two $H_2O$ vacancies. The simulation is done in the NVT ensemble (1600 K and a corresponding pressure of 110 GPa) with a time step of 1 fs. During the duration of this long simulation (97.5 ps of statistics after 2.5 ps of equilibration), we observe a total of 17 jumps, enough to obtain a linear regime in the mean square displacement (see Fig. S16) and a diffusion coefficient $D_O = 3.5 \times 10^{-12}$ m$^2$.s$^{-1}$. Given that in natural ice the concentration of vacancies may be lower than 2/128, our estimation of $D_O$, and thus $D$, likely represents an upper bound.

**Geodynamical scalings.** We use scaling laws obtained in the context of the dynamics of the Earth mantle to estimate the feasibility of solute transport by convection in the high-pressure ice layer from the rocky core of icy exoplanets to the overlying ocean. The first important control dimensionless number for this process, the Rayleigh number (Eq. 1), depends on several ill-constrained and planet-dependent parameters. Some physical parameters for bcc ices are obtained directly from the DFT-MD calculations ($\rho = 2.0 \times 10^3$ kg m$^{-3}$, $\alpha = 2 \times 10^{-4}$ K$^{-1}$), some are educated guesses based on constraints on ice VI ($\eta = 1.310^{17}$ Pas, $\kappa = 4.310^{-7}$ m$^2$s$^{-1}$, $k = 1.6$ Wm$^{-1}$ K$^{-1}$). The gravitational acceleration is computed as $g = 4\pi G\rho_c R_c/3$ with $G = 6.67 \times 10^{-11}$ m$^3$kg$^{-1}$s$^{-2}$ the gravitational constant. Conservative reference values for the planet-dependent parameters are used and the most uncertain or variable parameters are emphasized on the right-hand-side of Eq. 1.

The second important dimensionless control number is the buoyancy number which balances the density change from composition and that from temperature. Thermal convection in the ice layer proceeds by diffusive growth of a boundary layer until its thickness $\delta_T \sim \sqrt{\kappa t}$ reaches a critical value, set by the value of the boundary layer Rayleigh number $Ra_\delta = Ra(\delta_T/d)^4$ reaching the critical value for the onset of convection. This provides a critical time, $t_\star$. In the same time, the boundary layer gets enriched in salts by diffusion, on a thickness $\delta_\chi \sim \sqrt{Dt_\star}$, with a saturated value $\delta\rho_m = 7 \times 10^{-3}\rho$ at the rock-ice boundary. The temperature and composition profiles in the boundary layer take error function shapes with their respective thicknesses as scaling parameter but can be linearized to estimate that the total mass of salts in the composition boundary layer is $\delta\rho_m\sqrt{Dt_\star}/2$ per unit surface, which amounts to an average density increase in the thermal boundary layer from salt enrichment $\Delta\rho_\chi = \delta\rho_m\sqrt{D/\kappa}/2$. The value of $D$ was estimated from DFT-MD as detailed in the previous section. To estimate the temperature difference $\Delta T$ across the boundary layer, we use the scaling of the dimensionless heat flux as function of the Rayleigh number[46] as $Nu = 0.37Ra^{1/3}$ which provides a scaling for the thickness of the boundary layer as $d_{BL} = d/(0.37Ra^{1/3})$. The temperature difference is then $\Delta T = q_c d_{BL}/k$. Collecting these relationships gives for the buoyancy number

$$B = \frac{\delta\rho_m}{2\rho}\sqrt{\frac{D}{\kappa}}\frac{0.37Ra^{1/3}k}{dq_c\alpha} \quad (4)$$

Combining this equation with the scaling for the Rayleigh number (Eq. 1) leads to Eq. 2. It shows that the density increase from enrichment in salts is not likely to affect convection in the ice layer, as long as the transport at the interface happens by diffusion. If melting of ice occurs, a much more efficient transport process is possible[39–41].

## Data availability

The data generated in this study are provided in the article, in Supplementary Information, and in a public repository at https://osf.io/w64fm/ together with a numerical implementation of the free energy surface of NaCl-bearing bcc ice. Raw data from atomistic simulations are available from the corresponding author upon request.

## Code availability

DFT simulations have been performed with the proprietary code VASP (https://www.vasp.at/).

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

## Acknowledgements

We acknowledge access to the GENCI supercomputing centers (CINES, IDRIS, CCRT) through a series of eDARI grants for performing the simulations (stl2876). Additional computational resources were provided by the Norwegian infrastructure for high-performance computing (NOTUR grants NN9329K, NN2916K, and NN9697K). R.C. was funded by the European Research Council (ERC) under the European Union's Horizon 2020 research and innovation program (grant agreement n°681818 – IMPACT) and by the Deep Carbon Observatory under a grant from the Extreme Physics and Chemistry Directorate. J.A.H. and R.C. were funded by the Research Council of Norway through its Centres of Excellence funding scheme, project number 223272. SL was funded by a grant from the Agence Nationale de la Recherche (ANR) under project number ANR-15-CE31-0018-01.

## Author contributions

J.-A.H., R.C. and S.L. conceived the study and wrote the manuscript. J.-A.H. performed the atomistic simulations and analysis, J.-A.H and S.L. performed the geodynamic calculations.

## Competing interests

The authors declare no competing interests.
