## [Peer Review File · Nature Communications]

**REVIEWER COMMENTS**

**Reviewer #1 (Remarks to the Author):**

**In the present manuscript, Hernandez and co-authors are reporting new DFT-MD calculations on the**
**incorporation of sodium chloride into ice VII and ice X over a wide range of pressures, temperatures and**
**composition. Then, they discuss the potential implications for the vertical transport of nutrients in high pressure**
**ice mantles of water-rich planets, and their potential habitability.**

**The DFT-MD results are very well described and represent the most comprehensive and significant result to**
**date on the physics of salty high pressure ice. I have no major comment on the DFT and thermodynamic part as**
**the quality of the work is remarkable. The supplementary materials are well organized and provide all the**
**required details.**

**Nonetheless, I found that the manuscript is not reaching the threshold required for Nature communications.**
**Some of the major outcomes described here (shift in phase transition, influence of incorporation on the density,**
**etc.), have already been described in several prior studies (experimental and theoretical) on cubic HP ice with**
**various other salts. Even if the present work represent a significant step forward in our knowledge of salty HP**
**ices, it is not quite paradigm-shifting. There is also significant issues with the planetary implications section**
**described hereafter. Overall, the scope and quality of the manuscript qualify it more for publication in a more**
**specialized journal (e.g. Nature Materials) rather than Nature communications.**

**Major comments:**

**The geodynamic section of this manuscript ill-conceived. The analysis presented here is too simplistic, specially**
**when compared to the seriousness and quality of the DFT-MD results. Overall, it overlooks the complexity of**
**multi-phase convecting mantles with pressure and chemical dependent properties. This makes this section too**
**insubstantial to support appropriately the claims of the authors regarding the habitability of large ocean**
**planets. Furthermore, recent papers have made similar claims regarding the exchange of chemicals and**
**potential habitability of the ocean (Fu et al. 2010, Noack et al. 2017, Levi and Sasselov 2018), with more advance**
**models.**

**Incorporation of salts would create singular behavior (e.g. delayed phase transitions VII-X, change in solubility,**
**chemically driven density gradients) that cannot be taken into account by a simple analytical solution. For**
**example, it seems also clear that denser salty ice VII/X forming at the base of the high pressure ice mantle could**
**affect convection regimes as the chemical/density gradient would tend to fight against convection. This could**
**possibly form a thermo-chemical boundary layer at the foot of the adiabatic profile, where dense salty ice could**
**be trapped at the bottom, limiting/stopping the chemical exchanges with the ocean.**

**Arguably, the study of vertical transport of chemical in high-pressure ice mantles and habitability of the ocean**
**cannot be addressed with a simple analytical model. More advance geodynamic simulations are required to take**
**into account density differences between regions of different ices salinity, differences in thermodynamic**
**properties, solubility, phase transitions between ice polymorphs, effect of different exoplanet**
**size/heatflux/surface temperatures, etc.**

**Another major issue is that Choblet et al. (2017) never reported any parameters for ice VII transport**
**properties, but only for ice VI, contrary to what the authors claims. Ice VII creep properties are still**
**unconstrained experimentally or theoretically. Using ice VI for the calculation of the dimensionless Rayleigh**
**number makes this all section unsound unfortunately. No ice VII/X creep laws exists to date, and HP ices**
**viscosities can change significantly with the type of polymorph (Durham et al. 2010) and pressure, so using ice**
**VI, a tetragonal phase stable <2GPa, to describe cubic ice VII or X is not appropriate.**

Considering the quality and importance of the DFT-MD work, some of these aspects could be addressed with a more advance model, but this is most probably beyond the scope of the present work.

Therefore I would recommend to submit a reworked version of this manuscript focusing more on the DFT-MD results to another specialized journal. I hope my comments will help to get this very good manuscript published in a more appropriate journal as fast as possible.

Other comments:

Phase boundaries of the salty ice (starting 1.58)
the authors should report that the shift in the ice VII-X transition to higher pressure by incorporation of salt has been already reported by experimental and MD data for LiCl (Bove et al., 2015)

Figure 2: I would ask to add a legend for the colored/NaCl concentration for a,b and c sub-plots (as it is the case in the supplementary), this would greatly help the readability of the figure.

Figure 2d: Correct H₂O-6NaCl by NaCl-6H₂O

It would be nice if a numerical implementation of the Gibbs energy was provided as supplementary to allow the community to use this new representation directly "out of the box".

References:

Bove, L.E., Gaal, R., Raza, Z., Ludl, A.-A., Klotz, S., Saitta, A.M., Goncharov, A.F., Gillet, P., 2015. Effect of salt on the H-bond symmetrization in ice. PNAS 201502438. <https://doi.org/10.1073/pnas.1502438112>

Fu, R., O’Connell, R., Sasselov, D., 2010. The Interior Dynamics of Water Planets. The Astrophysical Journal 708, 1326–1334. <https://doi.org/10.1088/0004-637X/708/2/1326>

Levi, A., Sasselov, D., 2018. A New Desalination Pump Helps Define the pH of Ocean Worlds. ApJ 857, 65. <https://doi.org/10.3847/1538-4357/aab715>

Durham, W.B., Prieto-Ballesteros, O., Goldsby, D.L., Kargel, J.S., 2010. Rheological and Thermal Properties of Icy Materials. Space Sci Rev 153, 273–298. <https://doi.org/10.1007/s11214-009-9619-1>

Reviewer #2 (Remarks to the Author):

The manuscript by Hernandez et al. describes new findings regarding the maximum solubility of salts in high-pressure ice, which is an important information for exoplanet modellers studying the interior evolution of water-rich, massive exoplanets with high-pressure ice forming above a rocky core. I am not a mineral physicist, and I will comment mainly on the implications of the results of the study. I have no general, main points, but noticed a few more specific points when reading though the manuscript.

1) The phase boundaries section seems a little bit too technical for a broad audience as expected for a Nature

**Communications publication. Maybe it would be better to discuss details on different bonds in an**
**appendix/supplementary materials section; or alternatively summarize in plain language what the bond length**
**variations mean, i.e. what is their relevance? But this may be due to my missing background in mineral physics.**

**2) It would be advantageous if Figure 2 could have a legend showing how colours change with R, as I assume**
**that a, b and c all use the same colour scale. For panel d, is the colorbar title "Arbitrary units" really**
**appropriate here? Why not just write "Extrema" here?**

**3) It is not quite clear to me where the reformulated Rayleigh number calculation (1) comes from. This**
**conversion apparently does not come from the cited Choblet paper. Using a reference Rayleigh number (10^7)**
**for reference heat flux and mantle thickness seems reasonable, but why do you scale linearly with R_c ? Did you**
**calculate that the different material properties ($\alpha \cdot \rho \cdot g / (k \cdot \kappa \cdot \eta)$) scale linearly with core radius?**
**Should they not rather depend on the ice shell thickness above and more generally on planet mass?**

**Maybe one can use a simple rule-of-thumb approximation depending on planet radius/mass (and not core**
**radius):**

**If $R_p \sim M^{0.267}$ (Wagner et al. 2011, Icarus, for super-Ganymedes), then $g \sim (M/R_p^2) \sim M^{0.5}$;**

**$\rho \sim (M/R_p^3) \sim M^{0.2}$ and $\alpha/k/\kappa$ should not vary too strongly with mass, so your R_a should scale**
**rather with $\sim M^{0.7}$?**

**On the other hand, it seems the R_a number is actually not further used, but only needed to argue for large-**
**enough numbers to allow for convection in the ice shell.**

**4) Also, you mention that you use $\rho_c = 5000 \text{ kg/m}^3$ to compute gravity (I guess you calculate M from a given**
**R_c , and therefore need ρ_c ?), but where do you need the gravity?**

**Same for the convective velocity - you do give the equation, but not what numbers you would get for planets of**
**different size.**

**5) The paper ends rather abruptly after defining the different equations/relationships for convection, it seems**
**like a further application was planned here (and would be somewhat expected since you mention the water-rich**
**super Earths, so a quantitative evaluation would be useful), but instead it is only mentioned that convection**
**should occur and would lead to enrichment of electrolytes in the oceans, without a direct context to your results**
**on salt solubility. One could discuss for example that an increased density on the one hand leads as stated in**
**Figure 4 rather to sinking of salt intrusions; but on the other hand salty ice would also have a higher Rayleigh**
**number due to the density term - and there may be other effects on R_a , for example I would assume that the ice**
**conductivity would vary with salt content, and also the viscosity...**

**6) One question that may need to be discussed is also if during planet formation, when water and rocky core**
**separate (before cooling and adding of additional mass leads to high-pressure ice formation), if salts will not**
**anyway be washed out of the rocky core already and accumulate in the oceans in the beginning, and when ice VI**
**forms this would lead to a separation into pure water HP-ice and even higher salt concentrations in the oceans?**
**And later impacted material would also first make its way through the ocean, so maybe the salts would again**
**stay in the ocean? But this is maybe too speculative for this paper...**

**7) The abstract ends with "The presence of salt impurities enhances the diffusion of H atoms, extending the**
**stability field of recently discovered superionic ice, and pushes the transition to ice X towards higher pressures.**
**These findings suggest that the high-pressure ice mantle of water-rich super-Earths is permeable to the**
**transport of electrolytes between the inner rocky core and the outer liquid layer; such upward flux being**
**necessary to support life into shallow oceans."**

**I may have missed this - but where is this discussed and explained in the paper? Why does it matter where the**
**transition to ice X takes place?**

**I would like to add that I very much appreciate the detailed descriptions in the supplementary material. It is**
**clear that the authors put a lot of effort in it, and it is almost a shame that it is hidden in the SM and will not be**

seen be the average reader of the paper. Since the manuscript is not yet exhausting long, some of the material
from the SM may also fit in the actual manuscript?

**Reviewer #3 (Remarks to the Author):**

**The study uses first-principles simulations to demonstrate that up to 2.5 wt% NaCl can be dissolved in the very**
**high-pressure (HP) ices that may be present in large icy exoplanets, named icy super-Earths in this paper. Such**
**planets have a large fraction of H₂O, including a HP ice layer between the ocean and the silicate core. They infer**
**that electrolytes can be transported from the rocky inner core to the ocean. The paper concludes that this**
**finding is important because such elements can support life that may be present in shallow oceans.**

**The research is innovative as it applies state of the art first principles calculations to the fascinating domain of**
**exoplanets. However, I think that the paper would be improved if the authors can show some comparisons**
**between laboratory measurements and the theory described in this paper. Also, I think that the interior model**
**needs some improvements (see my comments below).**

**The supplementary information provides the descriptions of the thermodynamic quantities that are required to**
**compute the Gibbs free energies of a mixture of ice and NaCl. I haven't looked in the details of the 23 pages but**
**this seems a very solid work. My only comment is whether it would be possible to test the simulations by**
**comparing the melting curves of pure ice X obtained in lab and with these calculations. I was surprised that the**
**paper by Schwager and Boehler (2008) on ice X was not cited.**

**Also, there are some laboratory experiments on iver VII with NaCl and it would be good to show the comparison**
**between the theory and the laboratory measurements.**

**The application to life on exoplanets is a stretch. Maybe habitability rather than life may be a more reasonable**
**start of the paper as we do not even know for places like Europa and Enceladus whether life exists in their**
**oceans although their habitability has been demonstrated. What this paper focuses on is the composition of the**
**ocean and the ability of HP ices to carry electrolytes. The authors should consider revising the 3 first lines of**
**their abstract.**

**The authors consider icy super-Earths, which I interpret as large ocean worlds (Leger et al, 2004) covered by an**
**ice shell because the surface temperature is below the freezing point of water. Have such exoplanets been**
**discovered? From the catalog of 4100 confirmed exoplanets, how many could be in that category? The present**
**results can also apply to ocean worlds, relaxing a bit the constrain on surface temperature.**

**Figure 4: First, it would be useful to have a temperature profile of convection in the phase diagram of ice (ice**
**VII and ice X from Schwager and Boehler (2008)). If there is indeed an ice shell on top, then the adiabat in the**
**ocean should cross ice VI before crossing ice VII. This has implication since ice VI does not include NaCl. So ice**
**VI may act as a barrier to the transfer. It would be good to show the thermal profile. If the presence of ice on**
**top is relaxed, then the adiabatic gradient in the ocean may cross the ice VII and not the ice VI freezing curve.**
**This should be discussed in the paper.**

**The ocean is likely to have included Na and Cl during accretion when silicates could easily react with water as**
**the interior differentiated.**

**Most of the exoplanets have very large surface temperature. The authors should mention which exoplanets**
**could have such a low surface temperature that would allow ice to form.**

**Is ice VII* the new phase discovered by Schwager and Boehler (2008)?**

**Line 150: the presence of melt at the interface between HP ice and silicates depends on the efficiency of**
**convection in the ice X and ice VII layers. This was demonstrated by Kalousova and Sotin (2018, 2020) in the**

**case of the ice VI/silicates interface inside Ganymede and Titan.**

Point-by-point response to the reviewers

First of all, we would like to thank all three reviewers for their time and their comments on our manuscript.

We revised our manuscript according to the suggestions of the reviewers. In particular, we largely toned down the implications of our work on the potential habitability of large water-rich exoplanets as it is a complex multi-factorial question that goes beyond the scope of the present study. The new version of the manuscript focuses primarily on the DFT-MD results. Additional context and details on the different water ice phases have been added to the introduction, and to the section Properties and phase diagram of salty ice. These changes should make the DFT-MD work more accessible to the general reader. Finally, we decided to keep and to enrich the discussion on the effect of salt inclusion on the transport of electrolytes in large icy planets, considering different scenarii. As mentioned by the reviewers, an analytical convection model cannot describe all complex phenomena that would occur in high-pressure ice mantles. However, we think that it provides some interesting considerations and opens the path towards more complete numerical models. These analytical considerations have been enriched in the revised version of the manuscript and its limitations are explicitly mentioned. An additional DFT-MD simulation was run in order to estimate the diffusivity of NaCl (D) in the ice that is required in the expression of the buoyancy number. The corresponding details have been added in the Methods section and in Supplementary Material.

A revised version of the manuscript with all changes colored in blue can be found at the end of this document.

Reviewer #1

In the present manuscript, Hernandez and co-authors are reporting new DFT-MD calculations on the incorporation of sodium chloride into ice VII and ice X over a wide range of pressures, temperatures and composition. Then, they discuss the potential implications for the vertical transport of nutrients in high pressure ice mantles of water-rich planets, and their potential habitability.

The DFT-MD results are very well described and represent the most comprehensive and significant result to date on the physics of salty high pressure ice. I have no major comment on the DFT and thermodynamic part as the quality of the work is remarkable. The supplementary materials are well organized and provide all the required details.

Nonetheless, I found that the manuscript is not reaching the threshold required for Nature communications. Some of the major outcomes described here (shift in phase transition, influence of incorporation on the density, etc.), have already been described in several prior studies (experimental and theoretical) on cubic HP ice with various other salts. Even if the present work represents a significant step forward in our knowledge of salty HP ices, it is not quite paradigm-shifting. There are also significant issues with the planetary implications section described hereafter. Overall, the scope and quality of the manuscript qualify it more for publication in a more specialized journal (e.g. Nature Materials) rather than Nature communications.

We thank Reviewer #1 for recognizing the quality of our work. Based on the editors recommendations we submit a revised version of the manuscript to Nature Communications. We would like to underline that the influence of salt inclusion on the superionicity has never been shown.

To our opinion this particular aspect is paradigm-shifting as impurities are likely to be present in planetary ices, lowering the pressure and temperature threshold for the superionic state and its associated onset of electrical conductivity.

Major comments:

The geodynamic section of this manuscript ill-conceived. The analysis presented here is too simplistic, specially when compared to the seriousness and quality of the DFT-MD results. Overall, it overlooks the complexity of multi-phase convecting mantles with pressure and chemical dependent properties. This makes this section too insubstantial to support appropriately the claims of the authors regarding the habitability of large ocean planets. Furthermore, recent papers have made similar claims regarding the exchange of chemicals and potential habitability of the ocean (Fu et al. 2010, Noack et al. 2017, Levi and Sasselov 2018), with more advance models.

R1-1) Incorporation of salts would create singular behavior (e.g. delayed phase transitions VII-X, change in solubility, chemically driven density gradients) that cannot be taken into account by a simple analytical solution. For example, it seems also clear that denser salty ice VII/X forming at the base of the high pressure ice mantle could affect convection regimes as the chemical/density gradient would tend to fight against convection. This could possibly form a thermo-chemical boundary layer at the foot of the adiabatic profile, where dense salty ice could be trapped at the bottom, limiting/stopping the chemical exchanges with the ocean.

We thank the reviewer for pointing that out. We now address this issue in the revised paper. Although we agree that a large flux of salts at the base of the ice layer would ultimately counteract thermal convection, the much smaller diffusivity of salts in the ice compared to the estimated thermal conductivity makes it unlikely to happen. This means that a thermal boundary layer grows faster than a composition boundary layer and becomes unstable before the concentration of salts reaches sufficient levels to significantly affect the density of ice. In fact, we expect that salts would be transported efficiently to the overlying ocean at a rate larger than the one at which a large concentration can build up at the bottom of the ice layer. Clearly, this issue could be tested by complete numerical models but that falls out of the scope of the present paper.

R1-2) Arguably, the study of vertical transport of chemical in high-pressure ice mantles and habitability of the ocean cannot be addressed with a simple analytical model. More advance geodynamic simulations are required to take into account density differences between regions of different ices salinity, differences in thermodynamic properties, solubility, phase transitions between ice polymorphs, effect of different exoplanet size/heatflux/surface temperatures, etc.

The history of fluid dynamics shows very well that simple analytic work based on scaling analysis allows to decipher first-order phenomena (e.g. Scaling self-similarity and intermediate asymptotics, by Barenblatt, published by Cambridge Univ. Press, 1979). Without denying the usefulness of complex models as mentioned by the reviewer, the ill-constrained nature of most parameters makes, in our opinion, scaling approaches more useful at this stage, in particular since it allows researchers to reduce the parameter space to the least number of independent parameters. We consider only two such parameters, the Rayleigh and the buoyancy numbers, which are the most important adimensional parameters when discussing the possibility of convection and its efficiency. We use results from models that include the effects of the solid-liquid phase change (Agrusta et al, 2020), effects that have been shown to be quite drastic and have not been included in the more advanced models mentioned by the reviewer.

R1-3) Another major issue is that Choblet et al. (2017) never reported any parameters for ice

VII transport properties, but only for ice VI, contrary to what the authors claims. Ice VII creep properties are still unconstrained experimentally or theoretically. Using ice VI for the calculation of the dimensionless Rayleigh number makes this all section unsound unfortunately. No ice VII/X creep laws exists to date, and HP ices viscosities can change significantly with the type of polymorph (Durham et al. 2010) and pressure, so using ice VI, a tetragonal phase stable <2GPa, to describe cubic ice VII or X is not appropriate.

Mentionning ice VII in the previous version was a mistake and we thank the reviewer for catching that. Clearly, transport properties of bcc ices are unconstrained and we use values from ice VI only as a guide and keep the value on the right-hand-side of our equations so that readers can easily see the effects of changing these values. This shows that changes of several orders of magnitude in the viscosity leave the ice layer in a convecting state.

R1-4) Considering the quality and importance of the DFT-MD work, some of these aspects could be addressed with a more advance model, but this is most probably beyond the scope of the present work.

More advanced models are feasible and desirable in the future but fall beyond the scope of the present paper. We hope that this paper will be an incentive to go toward such models.

Other comments:

R1-5) Phase boundaries of the salty ice (starting 1.58)
the authors should report that the shift in the ice VII-X transition to higher pressure by incorporation of salt has been already reported by experimental and MD data for LiCl (Bove et al., 2015)

This was reported in the introduction of the manuscript (ref. 11 was Bove et al., 2015): The inclusion of salt decreases the melting temperature and breaks the connectivity of the hydrogen-bond network in solid ice. In ice VII the presence of halides increases the rotational disorder of water molecules, decreases the nuclear quantum effects associated to the H atoms¹⁵ and shifts upwards the symmetrization of the hydrogen bonds (the ice VII to ice X transition^{16, 17}) by about 40 GPa at ambient temperatures¹¹.

In the new version of the manuscript, this paragraph has been slightly modified: "The inclusion of salt in ice VII decreases its melting temperature (Journaux et al. 2013) and breaks the connectivity of its HB network (Klotz et al 2009, Ludl et al. 2017). In particular, the presence of halides increases the rotational disorder of water molecules and decreases the role of NQEs in the O–H···O bond symmetrization (Bronstein et al. 2016), which shifts the transition to ice X by about 40 GPa at ambient temperature (Bove et al. 2015).".

R1-6) Figure 2: I would ask to add a legend for the colored/NaCl concentration for a,b and c sub-plots (as it is the case in the supplementary), this would greatly help the readability of the figure.

We added a legend to Figure 2 for a., b., c. subplots. Moreover, we noticed that several H₂O data points were missing in panels a. and b. (they were present in panel c.). They have been added to the new Figure 2. This does not change anything to the interpretation.

R1-7) Figure 2d: Correct H₂O-6NaCl by NaCl-6H₂O

This has been corrected. In addition, we changed the stereographic plots: we replaced the angles by the crystallographic orientations, we changed the color scale and added a white contour corresponding to a given density of O-H bond orientation in order to make more evident the difference between the pure H₂O and NaCl·6H₂O cases. We also renamed the colorbar scale as Density of O-H bond orientations.

R1-8) It would be nice if a numerical implementation of the Gibbs energy was provided as supplementary to allow the community to use this new representation directly "out of the box".

As suggested by Reviewer #1, we now provide a numerical implementation of the thermodynamic potential representing the inclusion of NaCl in bcc ice at 1600 K in the VII'-X regimes. The corresponding Python scripts compute the thermodynamic potential on a given (V, w) grid that is then re-sampled in a given (p, w) grid in order to compute $\Delta g_{mix}(p, w)$. As already mentioned in the Supplementary Material, the model is valid for $w = [0; 0.19]$ wt% NaCl and $V = [14.75; 22.55]$ Å³.uc⁻¹. It can be found at <https://osf.io/w64fm/>. A README.txt provides a description of the files and explains how to generate data on a given grid. The model generated is also compared to the DFT-MD data that were used to fit it.

We added the link towards the code in the Method section of the revised manuscript and in the Supplementary Material.

Reviewer #2

The manuscript by Hernandez et al. describes new findings regarding the maximum solubility of salts in high-pressure ice, which is an important information for exoplanet modellers studying the interior evolution of water-rich, massive exoplanets with high-pressure ice forming above a rocky core. I am not a mineral physicist, and I will comment mainly on the implications of the results of the study. I have no general, main points, but noticed a few more specific points when reading though the manuscript.

R2-1) The phase boundaries section seems a little bit too technical for a broad audience as expected for a Nature Communications publication. Maybe it would be better to discuss details on different bonds in an appendix/supplementary materials section; or alternatively summarize in plain language what the bond length variations mean, i.e. what is their relevance? But this may be due to my missing background in mineral physics.

We acknowledge that the section entitled "Phase boundaries of the salty ice" was too technical for a broad audience and that a paragraph describing the different bcc ices and their relations with the H-bonding regimes was missing.

In the revised manuscript, we added a descriptive paragraph about the different bcc H₂O ices and their properties in the introduction. In addition, we modified the original section, which is now entitled "Properties and phase boundaries of salty bcc ice". In the new form, it is more descriptive, with an introduction paragraph on the corresponding simulations, and with a summary of the effect of the salt inclusion on the ice properties and phase boundaries. In particular, we added a phase diagram (Figure 3) showing the shift of solid-solid boundaries towards high-pressures with increasing concentrations of NaCl.

These changes should clarify our analysis and the relations between the bonding regimes and the properties of ice and thus make the manuscript more accessible to a broader audience.

R2-2) It would be advantageous if Figure 2 could have a legend showing how colours change with R, as I assume that a, b and c all use the same colour scale. For panel d, is the colorbar title "Arbitrary units" really appropriate here? Why not just write "Extrema" here?

As mentioned in the response to reviewer #1, we modified Figure 2 according to reviewer #1 and #2 comments. We added a legend to Figure 2 for a., b., c. subplots. We noticed that several H₂O data points were missing in panels a. and b. (there are present in panel c.). They have been added to the new Figure 2. This does not change anything to the interpretation. In Figure 2.d., we replaced the stereographic plot's angles by the crystallographic orientations, we changed the color scale and added a white contour to a given density of O-H bond orientation to facilitate the comparison between the pure and the salty ice. We also renamed the colour scale as Density of O-H bond orientations.

R2-3) It is not quite clear to me where the reformulated Rayleigh number calculation (1) comes from. This conversion apparently does not come from the cited Choblet paper. Using a reference Rayleigh number (10^7) for reference heat flux and mantle thickness seems reasonable, but why do you scale linearly with Rc? Did you calculate that the different material properties ($\alpha \cdot \rho \cdot g / (k \cdot \kappa \cdot \eta)$) scale linearly with core radius? Should they not rather depend on the ice shell thickness above and more generally on planet mass?

Maybe one can use a simple rule-of-thumb approximation depending on planet radius/mass (and not core radius):

If $R_p \propto M^{0.267}$ (Wagner et al. 2011, Icarus, for super-Ganymedes), then $g \sim (M/R_p^2) \sim M^{0.5}$; $\rho \sim (M/R_p^3) \sim M^{0.2}$ and $\alpha/k/\kappa$ should not vary too strongly with mass, so your Ra should scale rather with $M^{0.7}$?

On the other hand, it seems the Ra number is actually not further used, but only needed to argue for large-enough numbers to allow for convection in the ice shell.

This expression is obtained from the equation 4 from Choblet et al (2017), which describes convection with an imposed heat flux. In order to obtain it we replace the imposed temperature difference ΔT in the classical Rayleigh number by $q d / k$. We added a section in the methods part where we explain how the values scale with the core radius. Essentially, all physical parameters are considered constant, except for the ones directly linked to the planet size and thickness of the ice layer. These latter parameters as well as the least constrained physical parameters are left on the right hand side of the equations to offer readers the possibility to simply test other preferred values. The Rayleigh number, in addition to allowing us to address the question of the feasibility of convection, is used to compute convective velocities based on the scaling law from Agrusta et al (2020).

R2-4) Also, you mention that you use $\rho_c = 5000 \text{ kg/m}^3$ to compute gravity (I guess you calculate M from a given Rc, and therefore need ρ_c ?), but where do you need the gravity? Same for the convective velocity - you do give the equation, but not what numbers you would get for planets of different size.

The gravity acceleration enters the calculation of Rayleigh number. We compute the convective velocity for the chosen reference values of the various parameters, $\sim 1 \text{ m/yr}$. Using the expressions we provide, the reader can easily use other numerical choices for the parameters.

R2-5) The paper ends rather abruptly after defining the different equations/relationships for convection, it seems like a further application was planned here (and would be somewhat expected since you mention the water-rich super Earths, so a quantitative evaluation would be useful), but instead it is only mentioned that convection should occur and would lead to enrichment of electrolytes in the oceans, without a direct context to your results on salt solubility. One could discuss for example that an increased density on the one hand leads as stated in Figure 4 rather to sinking

of salt intrusions; but on the other hand salty ice would also have a higher Rayleigh number due to the density term - and there may be other effects on Ra, for example I would assume that the ice conductivity would vary with salt content, and also the viscosity...

The main point we want to make here is that convection in HP ice with the high solubility of salts makes possible efficient transport of electrolytes to the ocean. We agree with the reviewer, and we added a discussion about the density increase due to the presence of salts; we conclude that it will not be sufficient to impede thermal convection. Further studies on more complex effects are beyond our scope at this stage but clearly deserve interest in the future.

R2-6) One question that may need to be discussed is also if during planet formation, when water and rocky core separate (before cooling and adding of additional mass leads to high-pressure ice formation), if salts will not anyway be washed out of the rocky core already and accumulate in the oceans in the beginning, and when ice VI forms this would lead to a separation into pure water HP-ice and even higher salt concentrations in the oceans? And later impacted material would also first make its way through the ocean, so maybe the salts would again stay in the ocean? But this is maybe too speculative for this paper...

It is indeed possible that the formation process of these planets results in a salty ocean, in which case there is no issue of transport from silicates for the question of habitability. What we want to stress here is that, in what is usually considered as the least favorable case, convection in the HP ice layer could still lead to transport of ice to the ocean.

We refer Reviewer #2 to our answer to Reviewer #3's points **R3-4)** and **R3-5)** for a more detailed discussion about the various conditions that we can expect in large icy exoplanets.

R2-7) The abstract ends with "The presence of salt impurities enhances the diffusion of H atoms, extending the stability field of recently discovered superionic ice, and pushes the transition to ice X towards higher pressures. These findings suggest that the high-pressure ice mantle of water-rich super-Earths is permeable to the transport of electrolytes between the inner rocky core and the outer liquid layer; such upward flux being necessary to support life into shallow oceans." I may have missed this - but where is this discussed and explained in the paper? Why does it matter where the transition to ice X takes place?

As pointed out by Reviewer #2 we should have been clearer with this point. The transitions from ice VII to VII' and ice X are not first order and therefore are not associated with discontinuities in thermodynamic variables. However, when the symmetrization of the hydrogen-bonds is complete (ice X), the elastic constant increase more rapidly as function of pressure and the ice is globally stiffer and would likely being associated with an increase in viscosity compared to molecular ice VII.

We modified the end of the abstract in the revised manuscript to add this aspect and to replace the implications regarding the habitability by those regarding the convection and the transport of electrolytes in the ice layer:

"[...]. The presence of salt impurities enhances the diffusion of H atoms, extending the stability field of recently discovered superionic ice, and pushes the transition to the stiffer ice X phase towards higher pressures. Scaling laws for thermo-compositional convection show that salts entering the high pressure ice layer can be readily transported across. These findings suggest that the high-pressure ice mantle of water-rich super-Earths is permeable to the convective transport of electrolytes between the inner rocky core and the outer liquid layer."

In addition, we added a new paragraph in the introduction describing the different phases of bcc ice and the consequences on their properties that are of interest for planetary interiors:

"These transitions strongly influence the compressibility, elasticity and transport properties

of bcc ice[18,22,23,26-29]. For instance, the transition between superionic ice VII'' and ice VII' is associated with a first-order transition, a dip in the elastic constants and a strong decrease of the electrical conductivity[18,23]. The completion the hydrogen bond symmetrization makes ice X stiffer than other bcc ices and preliminary investigation of its rheology predict a change of slip-system around 250 GPa[27]."

Finally, we added a similar paragraph at the end of the section "Properties and phase boundaries of the salty ice" to summarize the effect of the inclusion of salt in the ice: In summary, the inclusion of salt distorts the bcc sub-lattice, increases the orientational disorder and weakens the hydrogen-bond network of the ice. This shifts the superionic VII'' - VII' - X transition sequence towards higher pressure with increasing salt concentrations (see Figure 3). Therefore, at given pressure-temperature conditions, salty ice is less stiff, has a greater electrical conductivity and is likely to be less viscous than pure ice.

R2-8) I would like to add that I very much appreciate the detailed descriptions in the supplementary material. It is clear that the authors put a lot of effort in it, and it is almost a shame that it is hidden in the SM and will not be seen by the average reader of the paper. Since the manuscript is not yet exhausting long, some of the material from the SM may also fit in the actual manuscript?

Indeed, the supplementary material contains a lot of information but we think that it is too technical for the broad audience of Nature Communications. The interested reader can still look for the technical details or the coefficients of the various thermodynamic functions used to build the free energy model.

Reviewer #3

The study uses first-principles simulations to demonstrate that up to 2.5 wt% NaCl can be dissolved in the very high-pressure (HP) ices that may be present in large icy exoplanets, named icy super-Earths in this paper. Such planets have a large fraction of H₂O, including a HP ice layer between the ocean and the silicate core. They infer that electrolytes can be transported from the rocky inner core to the ocean. The paper concludes that this finding is important because such elements can support life that may be present in shallow oceans.

The research is innovative as it applies state of the art first principles calculations to the fascinating domain of exoplanets. However, I think that the paper would be improved if the authors can show some comparisons between laboratory measurements and the theory described in this paper. Also, I think that the interior model needs some improvements (see my comments below).

R3-1) The supplementary information provides the descriptions of the thermodynamic quantities that are required to compute the Gibbs free energies of a mixture of ice and NaCl. I haven't looked in the details of the 23 pages but this seems a very solid work. My only comment is whether it would be possible to test the simulations by comparing the melting curves of pure ice X obtained in lab and with these calculations. I was surprised that the paper by Schwager and Boehler (2008) on ice X was not cited.

Is ice VII* the new phase discovered by Schwager and Boehler (2008)?

Schwager and Boehler (2008) investigated phase transitions in high-pressure water ice using diamond anvil cell experiments coupled with a laser-heating system. The phase transitions were monitored by visual observation of a motion in the laser-speckle pattern and by observation of a temperature plateau when increasing the laser energy (for the solid-liquid transition only). They reported a solid-solid transition around 35 GPa and 1000 K and melting points up to 90 GPa and

2500 K. In the previous version of the manuscript, Schwager and Boehler (2008) was not included for three reasons: 1) we did not discuss the melting of water ice as we did it in a previous study dedicated to pure H₂O ice (see Hernandez et al., JCP 2018), because it is not the scope of the present work; 2) Schwager and Boehler (2008) do not provide structural constraints on the ice structure and therefore it was difficult to associate the reported solid-solid transition to one of the transition observed in pure ice. While our manuscript was in the review process, new XRD measurements in pure H₂O (Queyroux et al., PRL 2020) have detected the transition between superionic ice VII'' and ice VII' (see black square in Figure 3 of the new manuscript). These new study suggests that Schwager and Boehler (2008) actually measured the same transition. In the new version of the manuscript, we added a phase diagram (Figure 3) showing the influence of NaCl on the solid-solid phase boundaries in bcc ices. In this context, we think it is now appropriate to compare our data with both Schwager and Boehler (2008) and Queyroux et al. (2020).

Regarding agreement between DFT-MD simulations with experimental data. DFT-MD simulations using GGA PBE exchange-correlation functional have been shown to reproduce well the properties of high-pressure bcc water ice up to megabar pressures (Schwegler et al. 2008, French et al. 2015, French et al. 2016, Hernandez et al. 2016, Hernandez et al. 2018). In particular, Schwegler et al. (2008) computed the melting curve of bcc ice and found a very good agreement with the experimental melting points from Schwager et al. (2008). The only clearly limitation of our approach is at low temperature, where the inclusion of nuclear quantum effects in the dynamics is necessary to properly reproduce the ice VII-VII'-X transition sequence (Bronstein et al. 2014).

R3-2) Also, there are some laboratory experiments on ice VII with NaCl and it would be good to show the comparison between the theory and the laboratory measurements.

Indeed, Frank et al. 2006, 2008, 2013 report experimental equation of state data on ice VII with NaCl. However, they assume that NaCl enter the ice VII structure with Na⁺ occupying an interstitial fcc site and Cl⁻ replacing a H₂O molecule on a bcc site. Moreover, they find than salty ice has a smaller volume than pure ice. These results are in sharp contraction with other experimental studies (*e.g.* Bronstein et al. 2016) and recent measurements in natural ice VII (Tschauner et al. 2018) that find that the inclusion of NaCl or other salts always increases the volume of the ice compare to pure H₂O. These studies also suggest that both Na and Cl substitute water molecules on bcc sites. Moreover, it is unclear how much salt is actually included in the ice in Frank et al. experiments. Finally, Frank et al. 2006 and 2013 hypothesized that the inclusion of Na⁺ and Cl⁻ in bcc ice would lower the transition pressure to ice X. This is again conflicting with more recent Raman measurements and theoretical works (Bronstein 2016, in agreement with our study) that show that the inclusion of NaCl push the transition to ice X to high pressures due to the inhibition of nuclear quantum effects and due to the increasing in orientational disordering of the water molecules (as shown in our study).

In the manuscript, our numerical results are compared several times to experimental results, in particular regarding the inclusion mechanism of Na and Cl and also regarding the effect of NaCl inclusion on the ice density: "At these conditions, salty superionic ice always presents both a larger volume (by +0.41 % for 2.5 wt% NaCl and by +0.86 % for 5.88 wt% NaCl, both at 1600 K and 100 GPa) and a larger density than pure water ice (Figure 4.a). Such amounts are in agreement with recent experimental and theoretical works performed on the inclusion of various halides (NaCl, LiCl, MgCl₂, RbI) in ice VII at lower pressures and temperatures (Frank et al. 2006, Klotz et al. 2009, Bove et al. 2015, Journaux et al. 2017, Ludl et al. 2017). A solubility of 2.5 wt% is still consistent with the observation of temperature-induced exsolution of some NaCl above 500 K in NaCl-doped ice VII whose initial concentration was higher (Frank et al. 2008). The volume increase due to the addition of salt contrasts with the XRD measurements of Frank et al. (Frank et al. 2008) but agrees with all previously mentioned studies (Klotz et al. 2009, Bove et al. 2015,

Journaux et al. 2017, Ludl et al. 2017).”

Finally, in the new version of the manuscript, we added a phase diagram showing the influence of the salt inclusion on the solid-solid transition in bcc ice (Figure 3). This figure includes a comparison between experimental measurements of both the melting line and the VII''-VII' transition in pure H₂O from Schwager et al. (2008) and the new results from Queyroux et al. (2020).

R3-3) The application to life on exoplanets is a stretch. Maybe habitability rather than life may be a more reasonable start of the paper as we do not even know for places like Europa and Enceladus whether life exists in their oceans although their habitability has been demonstrated. What this paper focuses on is the composition of the ocean and the ability of HP ices to carry electrolytes. The authors should consider revising the 3 first lines of their abstract.

We agree with Reviewer #3's comment and as suggested by all reviewers, we completely remove the application of our work to life on exoplanets and also toned down the implications for their habitability. We removed the first and the last sentences of the abstract, where life was mentioned. While we still mention the relation between the habitability and the presence of electrolytes in the ocean in the first sentence, the rest of the abstract focuses more on our results and on the permeability of the HP ice mantle to electrolytes.

R3-4) The authors consider icy super-Earths, which I interpret as large ocean worlds (Leger et al, 2004) covered by an ice shell because the surface temperature is below the freezing point of water. Have such exoplanets been discovered? From the catalog of 4100 confirmed exoplanets, how many could be in that category? The present results can also apply to ocean worlds, relaxing a bit the constrain on surface temperature.

Figure 4: First, it would be useful to have a temperature profile of convection in the phase diagram of ice (ice VII and ice X from Schwager and Boehler (2008)). If there is indeed an ice shell on top, then the adiabat in the ocean should cross ice VI before crossing ice VII. This has implication since ice VI does not include NaCl. So ice VI may act as a barrier to the transfer. It would be good to show the thermal profile. If the presence of ice on top is relaxed, then the adiabatic gradient in the ocean may cross the ice VII and not the ice VI freezing curve. This should be discussed in the paper.

Based on The NASA Exoplanet Archive (<https://exoplanetarchive.ipac.caltech.edu/>) more than 50% of the confirmed exoplanets present radii ranging between those of the Earth ($1 R_{\oplus}$) and Neptune ($3.9 R_{\oplus}$) and most of those larger than $1.6 R_{\oplus}$ have bulk densities compatible with the presence of large amount of water (*e.g.* Zeng et al., PNAS 116 (2019); Mousis et al., The Astrophysical Journal Letters, 896:L22 (2020)).

In the present work, the term icy super-Earths refers to all large water-rich exoplanets potentially presenting a high-pressure ice mantle, without constrains on the surface temperature or on the presence of an atmosphere, etc. The properties of salty ice derived from our simulations are of interest for the full range of water-rich planets where bcc ices can form, from the hypothetical large ocean exoplanets as defined by Leger et al. (2004) to mini-Neptunes and Neptune-like exoplanets. Of course, the implications regarding the habitability apply mainly to large ocean exoplanets with a relatively thin atmosphere and/or a low surface temperature in order to obtain maximal temperatures 400-500 K at the bottom of the (sub-)surface ocean. Such case is depicted in Figure 5 of the manuscript, with an ocean/ice VII boundary as ice VI melts below 350 K. In cases of planets with thick atmospheres or with surface temperatures allowing liquid water to be stable, most of the ocean would reach a supercritical state, and the habitability would not be preserved. In such planets, the ocean would also be in contact with bcc ice (VII ou superionic VII'' depending on the temperature) given that the total amount of H₂O is large enough to reach

the liquid/solid transition. If the surface temperature is very high, the conditions in the water layer may never intersect the melting line. In the case of an exoplanet with a very low surface temperature without atmosphere, a layer of ice VI could cap the ice VII layer, and the phase change VII to VI in up-welling plumes could lead to ex-solving NaCl to make a poly-crystalline assemblage, much like Earth's mantle. The gross density of the assemblage should be similar to that of salt-doped ice VII and, according to the discussion surrounding equation 2 in the paper, should not prevent convection to proceed. If anything, the exothermic character of the VI→VII phase change helps convective flow, as has been well documented for upper mantle transitions in Earth's mantle. Salt would then be delivered as such to the overlying ocean.

However, due to the zoology of possible exoplanet interiors, it would be infeasible to include a meaningful phase diagram with all possible temperature profiles. Instead, we have included the previous discussion at the end of the revised manuscript in order to complement previous considerations.

R3-5) The ocean is likely to have included Na and Cl during accretion when silicates could easily react with water as the interior differentiated.

As pointed out by Reviewer #3, the hydrous alteration of silicates during the accretion could enrich the ocean in salt. This idea is associated to a formation scenario where the planet is formed in a global accretion event from an initial mixture of volatiles, rocks and metallic material. While this scenario is the more likely, a differentiated planet could also have a late input of volatile elements that condensed further away from the host star. In this second case, the hydrous alteration of silicates would occur to a much smaller extent than in the first scenario and the ice would have a lower concentration in salt.

However, given that the direction of the original salt flux is dependent on the planetary formation scenario, we now consider the general situation where both the ocean and the HP ice mantle exchange salt each other as imposed by the melting boundary conditions at the ocean/mantle interface. By convection, a thermal plume in the mantle would drag ice that is more enriched in salt up to the ocean/ice boundary and then melt, releasing salt in the ocean. A more diffuse return flow of salt into the mantle occurs during crystallization of the water at the bottom of the ocean. The actual concentration of salt in the newly crystallized ice depends on the concentration of salt in the ocean and on the partitioning coefficient between water and bcc ices.

R3-6) Most of the exoplanets have very large surface temperature. The authors should mention which exoplanets could have such a low surface temperature that would allow ice to form.

Based on The NASA Exoplanet Archive (<https://exoplanetarchive.ipac.caltech.edu/>), at least 17 exoplanets have both a surface temperature lower than 300 K and a radius between 1 and 3.9 R_{\oplus} . This number is indeed relatively small but it is mainly due to an observational bias of the transit method that detects mostly exoplanets with short period orbits, which are thus closer to their star, and thus more likely to be hotter. In our Solar System, the majority of the planets have surface temperatures lower than 300 K.

R3-7) Line 150: the presence of melt at the interface between HP ice and silicates depends on the efficiency of convection in the ice X and ice VII layers. This was demonstrated by Kalousova and Sotin (2018, 2020) in the case of the ice VI/silicates interface inside Ganymede and Titan.

Indeed and these important papers are cited in the revised version. What we show here is that even without melting we expect transport of salts to the ocean by subsolidus convection.

1 Solubility of salt in superionic ice suggests 2 electrolyte permeability in mantles of large icy 3 exoplanets

4 Jean-Alexis Hernandez^{1,2,*}, Razvan Caracas^{2,1}, and Stéphane Labrosse²

[revised manuscript text omitted]
^{12,44–46}. Based on the NASA Exoplanet Archive⁴⁷ more than 50% of the discovered exoplanets
 present radii ranging between those of the Earth ($1 R_\oplus$) and Neptune ($3.9 R_\oplus$) and most of those larger
 than $1.6 R_\oplus$ have bulk densities compatible with the presence of large amount of water^{48,49}. The properties
 of salty ice derived from our simulations are of interest for the full range of water-rich planets where bcc
 ices can form, from the hypothetical large ocean exoplanets as defined by Leger et al.⁵⁰ to mini-Neptunes
 and Neptune-like exoplanets. Of course, the implications regarding the habitability apply mainly to large

Figure 5. Transport of salt through the high-pressure ice mantle of a hypothetical large icy exoplanet. A thermal plume (dark orange) creates a focused upward flux of salty ice that melts at the boundary with the ocean. The crystallization at the bottom of the ocean over a broad area produces a diffuse return flow of salt in the mantle. Depending on the initial conditions and on the partitioning coefficients of NaCl between the ice and the ocean, the icy mantle may act either as a well or a source of electrolytes for the ocean.

ocean exoplanets with a relatively thin atmosphere and/or a low surface temperature in order to obtain
 maximal temperatures on the order of 400-500 K at the bottom of the (sub-)surface ocean. Such case
 is depicted in Figure 5, with an ocean/ice VII boundary as ice VI melts below 350 K. In other cases,
 for example of planets with thick atmospheres or with surface temperatures allowing liquid water to be
 stable, most of the ocean would reach a supercritical state, and the habitability would not be preserved. In
 such planets, the ocean would also be in contact with bcc ice (VII or superionic VII' depending on the
 temperature) given that the total amount of H₂O is large enough to reach the liquid/solid transition. If the
 surface temperature is very high, the conditions in the water layer may never intersect the melting line. In

the case of an exoplanet with a very low surface temperature without atmosphere, a layer of ice VI could
cap the ice VII layer, and the phase change VII to VI in up-welling plumes could lead to ex-solving NaCl
to make a polycrystalline assemblage. The gross density of the assemblage should be similar to that of
salt-doped ice VII and, according to the discussion surrounding equation 2, should not prevent convection
to proceed. If anything, the exothermic character of the VI→VII phase change helps convective flow, as
has been well documented for the phase transitions in the upper mantle of Earth’s mantle. Salt would then
be delivered as such to the overlying ocean.

Depending on the planetary formation scenario, the initial electrolyte concentration in the water-rich
envelopes may vary. If water is brought after the crystallization of the rocky mantle, water may remain
relatively free from alteration products. If the accretion of the icy and rocky materials is synchronous, the
water-based layers would likely contain a certain initial concentration of electrolytes due to the production
of salts by hydrous alteration of chondritic material⁶. In this case, the evolution of the ocean salinity would
depend on the partitioning of salts between liquid water and ice during crystallization at the bottom of
the ocean. The enrichment in salt of the icy mantle would be influenced by the nature of the water/rock
interface. In the case of a solid-solid interface, the enrichment of the ice layer depends on the partitioning
coefficients of each chemical species between the silicates and the cubic ice. If partial melting occurs at
the bottom of the high-pressure ice mantle due to the temperature increase in a thermal boundary layer,
the incorporation of impurities in the ice could be much more efficient than in the case of a solid-solid
interface^{36–38}. Future studies need to estimate the partitioning coefficients of essential electrolytes between
liquid water, superionic ice, and silicates at conditions comparable to the base of the ice layer in ocean
planets to improve our understanding of the internal functioning of these planets and to better assess their
habitability.

Methods

First-principles molecular dynamics

**Computational details.** We study the behavior of NaCl-bearing dense H₂O ice using first-principles
molecular dynamics (DFT-MD) simulations performed with the Vienna Ab initio Simulation Package^{51,52}
(VASP v4.6 and v5.2). To model the atoms, we used the PAW potentials^{53,54} with the following elec-
tronic configurations: O:[He] 2s²2p⁴, H:1s², Na:[Ne] 3s¹ and Na:[He] 2s²2p⁶3s¹, and Cl:[He] 2s²2p⁵. The
electronic structure was calculated within the density functional theory (DFT) framework and the Perdew-
Burke-Ernzerhof general gradient approximation⁵⁵ was used to approximate the exchange-correlation
term. Most DFT-MD simulations were performed with 4 × 4 × 4 bcc supercells (see Table 1) with a cutoff
energy of 550 eV for the plane-waves that ensured that internal energy is converged within 5 meV per
atoms, and 1.5 GPa precision in pressure with respect to a calculation with a cutoff energy of 1000 eV. We
calculated electronic wavefunctions at the Γ -point of the Monkhorst k-point grid⁵⁶. Time steps of 0.5 or
1 fs were employed in all simulations. DFT-MD have either been performed in the NVT, *i.e.* constant
Number of particles, Volume, and Temperature, and NPT, *i.e.* constant Number of particles, Pressure,
and Temperature, ensembles. For NVT sampling, temperature was controlled using velocity rescaling
(isokinetic) as no difference was observed for averaged properties when using a Nose-Hoover thermo-
stat^{57,58}. Isobaric-isothermal simulations were performed with Raman-Parrinello molecular dynamics⁵⁹
and a Langevin thermostat.

**Determination of the free energy of mixing.** The estimation of the thermodynamic stability of the
binary system requires the calculation of the Gibbs free energy of mixing $\Delta g_{mix}(w)$ as a function of salt
concentration w , pressure P , and temperature T . $\Delta g_{mix}(w, P, T)$ is related to the Gibbs free energies of the

pure phases ($g_{\text{H}_2\text{O}}(P, T)$, $g_{\text{NaCl}}(P, T)$) and to the configuration averaged Gibbs free energy of the mixture
 ($g_{\text{sol}}(w, P, T)$) as:

$$\Delta g_{\text{mix}}(w, P, T) = g_{\text{sol}}(w, P, T) - (wg_{\text{NaCl}}(P, T) + (1 - w)g_{\text{H}_2\text{O}}(P, T)) \quad (3)$$

[revised manuscript text omitted]

planets. *The Astrophys. J.* **784**, 96 (2014).
- **47.** NASA Exoplanet Archive. URL <https://exoplanetarchive.ipac.caltech.edu/>. The
NASA Exoplanet Archive is operated by the California Institute of Technology, under contract with
the National Aeronautics and Space Administration under the Exoplanet Exploration Program.
- **48.** Zeng, L. *et al.* Growth model interpretation of planet size distribution. *Proc. Natl. Acad. Sci.* **116**,
9723–9728 (2019).
- **49.** Mousis, O. *et al.* Irradiated ocean planets bridge super-earth and sub-neptune populations. *The*
*Astrophys. J.* **896**, L22 (2020). DOI 10.3847/2041-8213/ab9530.
- **50.** Léger, A. *et al.* A new family of planets? “ocean-planets”. *Icarus* **169**, 499–504 (2004).
- **51.** Kresse, G. & Furthmüller, J. Efficiency of ab-initio total energy calculations for metals and semicon-
ductors using a plane-wave basis set. *Comput. Mat. Sci.* **6** (1996).
- **52.** Kresse, G. & Furthmüller, J. Efficient iterative schemes for ab initio total-energy calculations using a
plane-wave basis set. *Phys. Rev. B* **54** (1996).
- **53.** Blöchl, P. E. Projector augmented-wave method. *Phys. Rev. B* **50** (1994).
- **54.** Kresse, G. & Joubert, D. From ultrasoft pseudopotentials to the projector augmented-wave method.
*Phys. Rev. B* **59**, 1758 (1999).
- **55.** Perdew, J. P., Burke, K. & Ernzerhof, M. Generalized gradient approximation made simple. *Phys.*
*Rev. Lett.* **77**, 3865–3868 (1996).
- **56.** Monkhorst, H. J. & Pack, J. D. Special points for brillouin-zone integrations. *Phys. review B* **13**, 5188
(1976).
- **57.** Nose, S. A unified formulation of the constant temperature molecular dynamics methods. *The J.*
*Chem. Phys.* **81** (1984).
- **58.** Hoover, W. G. Canonical dynamics: Equilibrium phase-space distributions. *Phys. Rev. A* **31** (1985).
- **59.** Parrinello, M. & Rahman, A. Polymorphic transitions in single crystals: A new molecular dynamics
method. *J. Appl. physics* **52**, 7182–7190 (1981).
- **60.** Lin, S.-T., Blanco, M. & III, W. A. G. The two-phase model for calculating thermodynamic properties
of liquids from molecular dynamics: Validation for the phase diagram of lennard-jones fluids. *The J.*
*Chem. Phys.* **119**, 11792–11805 (2003). DOI 10.1063/1.1624057.
- **61.** Lai, P.-K., Hsieh, C.-M. & Lin, S.-T. Rapid determination of entropy and free energy of mixtures
from molecular dynamics simulations with the two-phase thermodynamic model. *Phys. Chem. Chem.*
*Phys.* **14**, 15206–15213 (2012).

- **62.** Desjarlais, M. P. First-principles calculation of entropy for liquid metals. *Phys. Rev. E* **88**, 062145
(2013). DOI 10.1103/PhysRevE.88.062145.
- **63.** Grau-Crespo, R., Hamad, S., Catlow, C. & De Leeuw, N. Symmetry-adapted configurational modelling
of fractional site occupancy in solids. *J. Physics: Condens. Matter* **19**, 256201 (2007).

**Acknowledgements**

We acknowledge access to the GENCI supercomputing centers (CINES, IDRIS, CCRT) through a series of
eDARI grants for performing the simulations (stl2876). Additional computational resources were provided
by the Norwegian infrastructure for high-performance computing (NOTUR grants NN9329K, NN2916K,
and NN9697K). RC was funded by the European Research Council (ERC) under the European Union's
Horizon 2020 research and innovation program (grant agreement n°681818 – IMPACT) and by the Deep
Carbon Observatory under a grant from the Extreme Physics and Chemistry Directorate. JAH and RC were
funded by the Research Council of Norway through its Centres of Excellence funding scheme, project
number 223272. SL was funded by a grant from the Agence Nationale de la Recherche (ANR) under
project number ANR-15-CE31-0018-01.

**REVIEWER COMMENTS**

**Reviewer #1 (Remarks to the Author):**

**I thank the authors for the detailed response and their efforts to address our many comments.**

**The new analysis including the calculation of the buoyancy number is very useful and makes the claim from the**
**authors about the efficient chemical transport far more convincing. I also found the python script to compute**
**the thermodynamic potential very valuable. I thank the authors for making the effort to provide it.**

**I unfortunately found a major weakness in the present manuscript that further convince me that these DFT**
**results, as good as they are, cannot apply to water-rich super-earth, and that the habitability and planetary**
**implications are far more limited than I initially thought. There is other smaller comments that require some**
**change reported further.**

**Major comment:**

**About the planetary implications, the main issue is the fact that to reach the conditions of the calculation at**
**1600K in the high-pressure ice mantle (or super-ionic conditions for that matter), the surface temperature and**
**the amount of water need to be incredibly high. There is a general misconception that somehow the range**
**between super-Earth and mini Neptune is continuous and that any configuration of radius/composition are**
**possible in between. Careful statistical work on exoplanet catalogue and exoplanet formation show that this is**
**not the case and these are most probably two very distinctive populations with different formation history,**
**compositions, atmospheric thicknesses and habitability potential (Fulton et al. 2017, Lehmer et al. 2017).**

**From a simple calculations using a rough estimation of the adiabatic profile ($dT/dz = \alpha \cdot g \cdot T / C_p$) for water**
**and ice VII, one can see that a super-Earth planet (<1.8 Earth radius, 300-355K at the surface) would never**
**reach 1600K in the HP ices, whatever amount of water you put on top of it (the more water you put at constant**
**radius, the lower the g, the more gentle the thermal gradient). For larger mini-Neptune planets around 2.5**
**Earth radius (but probably with a very thick H/He atmosphere (Lehmer et al. 2017)), at least 30wt% H₂O are**
**necessary (and this is probably an exaggerated minimum, dense atmosphere will tend to make the observed**
**radius much larger than the condensed phases radius). Furthermore, this would only be the bottom condition,**
**with most of the HP ice mantle at temperature <1000K. Higher surface temperatures would allow higher HP ice**
**mantle T of course, but the surface /bottom temperature scales almost linearly. This means that to reach such**
**temperature in the HP ice mantle, the oceanic top conditions will be likely above the water critical point, not**
**compatible with habitability.**

**Actually most water-rich super-Earth (<1.8Earth Radius) with <30wt% H₂O and surface temperature < 355K**
**will have temperature at the bottom of their hydrosphere <800K and pressures <40GPa (below the ice X**
**transition), where super-ionicity conditions are not reached. A published example is given for Trappist 1f (1**
**Earth Radius and 50wt% H₂O, 220K surface T) in Unterborn et al. (2018), with an ice VII bottom temperature**
**of only 324K and a bottom pressure of only 25 GPa!**

**Therefore, the results presented here on super-ionic ice mostly apply to the mini-neptune/Neptune exoplanets**
**population with Radii > 2 Earth radius, (Fulton et al. 2017) and probably very thick H/He atmosphere, rather**
**than to potentially habitable water-rich super Earths (Lehmer et al. 2017).**

**As the authors state themselves l.179: "Of course, the implications regarding the habitability apply mainly to**
**large ocean exoplanets with a relatively thin atmosphere and/or a low surface temperature in order to obtain**
**maximal temperatures on the order of 400-500 K at the bottom of the (sub-)surface ocean. (...) In other cases,**
**for example of planets with thick atmospheres or with surface temperatures allowing liquid water to be stable,**
**most of the ocean would reach a supercritical state, and the habitability would not be preserved".**

**For this reason I think the implications for habitability and water-rich super Earth of the present work is**

unfortunately not quite credible. I would require the authors to remove any reference to habitability and
rework the implications (and title) of the current manuscript accordingly to only apply it to mini-
neptune/Neptune type exoplanets. If the authors chose to keep this aspect, it will be important to provide an
estimation (even through a rough adiabatic profile calculation) of the minimum planetary radius / water wt% /
surface temperature for which these results becomes applicable. This will be important for the planetary science
community so they can adequately grasp the relevance of the present work. I'm suspecting that only inhabitable
mini-neptunes with pretty high surface temperatures will allow such conditions to occur within more than 20%
of their hydrospheres.

**Other comments:**

**Add references to Fulton et al. (2017) and Lehmer et al. (2017) which are relevant to the Super-Earth/mini-
Neptune separation.**

**I'm surprised that the conductivity and diffusion results are not discussed in the main manuscript. These are
important and should be present in the main body, and not only in the supplementary.**

**P.7 regarding the value for ice VI used in equation 1. I would encourage the authors to point out that no ice VII
viscosity measurements exists to date. Hopefully this will motivate future experimental effort.**

**The last couple of sentences l.192 are incorrect and should be removed:**

**"If anything, the exothermic character of the VI→VII phase change helps convective flow, as has been well
documented for the phase transitions in the upper mantle of Earth's mantle. Salt would then be delivered as
such to the overlying ocean"**

**As suggested in Journaux et al. (2017), the exsolution of dense NaCl or NaCl hydrates at the VI-VII boundary
will make the salt tend to be gravitationally driven downwards.**

**The latent heat generation as an engine for convection claimed by the authors during the ice VII->VI reaction
seems difficult to justify. Indeed the VI-VII reaction is mostly isobaric, and result mainly in an volume change.
A very minor entropy/enthalpy change is to be expected from this transition, unlikely to provide any significant
thermal energy. Indeed, the VII->VI DeltaH is only 394 J/mol at 300K if you believe Dunaeva et al. (2010).
Furthermore the salty ice VII-> ice VI+NaCl hydrates latent heat budget remains totally unknown (it could be
endothermic), so I would just abstain form making any claims here.**

**The experimental observations reporting that salts are incompatible with ice VI needs to be more clearly
referenced. Add references to Journaux et al. (2017) at the end of the following sentences :**

**- l.17: "the presence of an underlying high-pressure ice mantle might prevent the transport of electrolytes due to
their insolubility in ice VI."**

**- l.190: "... ex-solving NaCl to make a polycrystalline assemblage."**

**Figure 5: There is a several issues with this figure, as pointed out in my major comment. If you have ice X and
ice VII' present, this cannot be a super-Earth, and the surface temperature cannot allow ice Ih to form. I would
remove the figure and replace it with one the DFT results from the supplementary material (e.g. Fig. S.14).**

**References:**

**Dunaeva, A.N., Antsyshkin, D.V., Kuskov, O.L., 2010. Phase diagram of H₂O: Thermodynamic functions of the
phase transitions of high-pressure ices. Solar System Research 44, 202–222.
<https://doi.org/10.1134/S0038094610030044>**

**Fulton, B.J., Petigura, E.A., Howard, A.W., Isaacson, H., Marcy, G.W., Cargile, P.A., Hebb, L., Weiss, L.M.,**

Johnson, J.A., Morton, T.D., Sinukoff, E., Crossfield, I.J.M., Hirsch, L.A., 2017. The California-Kepler Survey.
III. A Gap in the Radius Distribution of Small Planets. *The Astronomical Journal* 154, 109.
<https://doi.org/10.3847/1538-3881/aa80eb>

Lehmer, O.R., Catling, D.C., 2017. Rocky Worlds Limited to ~ 1.8 Earth Radii by Atmospheric Escape
during a Star's Extreme UV Saturation. *ApJ* 845, 130. <https://doi.org/10.3847/1538-4357/aa8137>

Unterborn, C.T., Desch, S.J., Hinkel, N.R., Lorenzo, A., 2018. Inward migration of the TRAPPIST-1 planets as
inferred from their water-rich compositions. *Nat Astron* 2, 297–302. <https://doi.org/10.1038/s41550-018-0411-6>

**Reviewer #2 (Remarks to the Author):**

**I would like to thank the authors for putting a lot of effort in revising the manuscript. I suggest that the study**
**can be published as it is.**

**I have only one last remark, based on the comments of the third reviewer - icy bodies for me also mean water-**
**rich planets covered by an ice shell, independent of if any HP ice would form. Since you mean the opposite here,**
**it would be good to either rephrase or explain directly in the abstract/introduction that the ice refers to the**
**bottom of the water layer and not the surface. But even better would be to rephrase the title of the manuscript**
**to not confuse the reader.**

Reviewer #4 (Remarks to the Author):

Solubility of salt in superionic ice suggests electrolyte permeability in mantles of large icy exoplanets

Jean-Alexis Hernandez, Razvan Caracas and Stephane Labrosse

The manuscript describes the incorporation of salt (NaCl) in high pressure ice structures (VII, VII', VII'' and X) along an isothermal path of 1600K using state of the art DFT calculations. The results are then incorporated in a convection model of an ice-giant planet to follow the effect of salt on the convection and on the transport of electrolytes through an icy mantle.

I am not an expert of DFT calculation but it seems the work presented here (also judging from the comments of other reviewers) is of remarkable quality, very well documented with all aspects described in the supplementary material.

The title only describes the solubility in superionic ices, but actually I found that the discussion concerns all ice structures investigated in the MS from ice VII to ice X, although at 1600 K they could be in the superionic state. Actually the first layer here to uptake salt from the rocky core is ice X, that may or may not be in this superionic state depending on pressure. The latter will depend on P-T conditions. The title seems restricted to a certain type of ice giant planet while the authors discuss a multi-layer ice mantle that opens a multitude of questions whether the conditions that are chosen for the convection model are the most common one. How is this establish that the best adiabat is at 1600 K? if the adiabat is slightly lower, how this influence the convection model (for instance if there is only on layer of superionic ice)? How is the thermal profile established here to obtain a convection?

Although, the different scenarii are somehow discussed, the case study concerns only one particular type of planet. This is where the mantle dynamic modelling is a bit weaker compared to the DFT part, and the latter could stand alone, and I would join reviewer 1 to say that the paper has a strong orientation on the physics of ices and the modelling could be in a separate study with more details and a variety of cases.

In the calculation they can incorporate quite an amount of salt in the structure because the liquid is quench to a crystal by sudden increase in pressure in 12 ps. Large amount of salt in ice was also incorporated experimentally by quenching in temperature a salty water then increasing the pressure and temperature to conduct neutron experiment in ice VII.

In both cases it seems there is quite of a trick to add large amount of salt but I am wondering how this would actually be the case when there is a solid-solid interface at the CMB of such planet?

What is the driving force to entrain electrolytes upward to the ocean in this case? It seems from the cartoon that the plume cannot be driven by a low density ice channel, because the salty ice is denser than the normal ice. So this is quite opposite to a rocky mantle where plumes are supposed to be hotter and less dense. In figure 5, it is referred as a thermal pile transport, so a thermal effect, but it is a competition between incorporation of salt (changing the density) and ΔT to make the ice lighter. Is there any evaluation of a threshold for both parameters (X_{NaCl} and T) to maintain the convection?

In such salty plumes how much is transferred to the rest of the icy mantle by diffusion and how much can be transferred to the next ice layer? Is there an estimate of the circulation time through the mantle layers?

Finally, I would point out that the modelling assumes an infinite reservoir of salt at the base of the mantle, but at such P-T conditions the rocky core should not react very much (or at least there is no data about partitioning between ice and rocks in such conditions) and once the elements of interest have been stripped away from a certain layer, how this is replenished? How can the flux of electrolyte keep constant in such model?

Overall this a strong and interesting contribution worthy of publication in Nature Communication.

Point-by-point response to the reviewers

First of all, we would like to thank the reviewers for the time they dedicated to our manuscript and for their constructive comments and suggestions. In the following, we answer in detail to all the points they raised.

In summary, we address the concerns of the reviewers relative to the application of our DFT results (showing the inclusion of relatively large amounts of salt in the structure of cubic ices) to water-rich super-Earths. In particular, we first avoid the confusion of the terminology used to describe water-rich exoplanets that present a high-pressure ice mantle. The term *icy super-Earths* was employed in the previous version as a generic name for water-rich exoplanets with a radius ranging between 1 and 4 Earth's radii (R_{\oplus}). This confused the reviewers, who were associating this term *icy* to a planet presenting ice at its surface, and the term *super-Earth* to planets with $R < 1.6 - 1.8R_{\oplus}$. In the new version of the manuscript, the term *super-Earth* is reserved to planets with $R < 1.6 - 1.8R_{\oplus}$ and the term *water-rich* is used instead of *icy*. Second, we clarify the objective of the analytical and general modelling done in section *Transport of electrolytes in a high-pressure ice mantle* and choose a better defined and realistic case for the numerical application: a planet of $1 M_{\oplus}$ (corresponding in this case to $\sim 1.3 R_{\oplus}$), composed 50 wt% H_2O and a Earth-like core, with no (or a very thin) atmosphere and a surface temperature creating 'habitable' conditions in surface liquid water layer (more detail below). We think that this section brings informative results and concepts that can serve as a basis for future studies based on more complex numerical models.

Reviewer #1

I thank the authors for the detailed response and their efforts to address our many comments.

The new analysis including the calculation of the buoyancy number is very useful and makes the claim from the authors about the efficient chemical transport far more convincing. I also found the python script to compute the thermodynamic potential very valuable. I thank the authors for making the effort to provide it.

I unfortunately found a major weakness in the present manuscript that further convince me that these DFT results, as good as they are, cannot apply to water-rich super-earth, and that the habitability and planetary implications are far more limited than I initially thought. There is other smaller comments that require some change reported further.

Major comment:

About the planetary implications, the main issue is the fact that to reach the conditions of the calculation at 1600K in the high-pressure ice mantle (or super-ionic conditions for that matter), the surface temperature and the amount of water need to be incredibly high. There is a general misconception that somehow the range between super-Earth and mini Neptune is continuous and that any configuration of radius/composition are possible in between. Careful statistical work on exoplanet catalogue and exoplanet formation show that this is not the case and these are most

probably two very distinctive populations with different formation history, compositions, atmospheric thicknesses and habitability potential (Fulton et al. 2017, Lehmer et al. 2017).

From a simple calculations using a rough estimation of the adiabatic profile ($dT/dz = \alpha * g * T / C_p$) for water and ice VII, one can see that a super-Earth planet (<1.8 Earth radius, 300-355K at the surface) would never reach 1600K in the HP ices, whatever amount of water you put on top of it (the more water you put at constant radius, the lower the g , the more gentle the thermal gradient). For larger mini-Neptune planets around 2.5 Earth radius (but probably with a very thick H/He atmosphere (Lehmer et al. 2017)), at least 30wt% H₂O are necessary (and this is probably an exaggerated minimum, dense atmosphere will tend to make the observed radius much larger than the condensed phases radius). Furthermore, this would only be the bottom condition, with most of the HP ice mantle at temperature <1000 K. Higher surface temperatures would allow higher HP ice mantle T of course, but the surface /bottom temperature scales almost linearly. This means that to reach such temperature in the HP ice mantle, the oceanic top conditions will be likely above the water critical point, not compatible with habitability.

Actually most water-rich super-Earth (<1.8 Earth Radius) with <30 wt% H₂O and surface temperature <355 K will have temperature at the bottom of their hydrosphere <800 K and pressures <40 GPa (below the ice X transition), where super-ionicity conditions are not reached. A published example is given for Trappist 1f (1 Earth Radius and 50wt% H₂O, 220K surface T) in Unterborn et al. (2018), with an ice VII bottom temperature of only 324K and a bottom pressure of only 25 GPa!

Therefore, the results presented here on super-ionic ice mostly apply to the mini-neptune/Neptune exoplanets population with Radii > 2 Earth radius, (Fulton et al. 2017) and probably very thick H/He atmosphere, rather than to potentially habitable water-rich super Earths (Lehmer et al. 2017).

As the authors state themselves 1.179: ” *Of course, the implications regarding the habitability apply mainly to large ocean exoplanets with a relatively thin atmosphere and/or a low surface temperature in order to obtain maximal temperatures on the order of 400-500 K at the bottom of the (sub-)surface ocean. (...) In other cases, for example of planets with thick atmospheres or with surface temperatures allowing liquid water to be stable, most of the ocean would reach a supercritical state, and the habitability would not be preserved*”.

For this reason I think the implications for habitability and water-rich super Earth of the present work is unfortunately not quite credible. I would require the authors to remove any reference to habitability and rework the implications (and title) of the current manuscript accordingly to only apply it to mini-neptune/Neptune type exoplanets. If the authors chose to keep this aspect, it will be important to provide an estimation (even through a rough adiabatic profile calculation) of the minimum planetary radius / water wt% / surface temperature for which these results becomes applicable. This will be important for the planetary science community so they can adequately grasp the relevance of the present work. I’m suspecting that only inhabitable mini-neptunes with pretty high surface temperatures will allow such conditions to occur within more than 20% of their hydrospheres.

We thank Reviewer #1 for raising this important point that allowed us to further improve the section relative to the planetary implications.

After carefully reading Fulton et al. (2017) and Lehmer et al. (2017), we agree that there are indeed two distinctive populations of planets that emerge from the exoplanet catalogue when sorting them by their radii, as shown by the figure below (from Lehmer et al. 2017):

The population density presents a minimum at 1.6-1.8 Earth radius that is interpreted as a transition between terrestrial planets (including super-Earths) and gas giants (including so-called mini-Neptunes). Two explanations have been proposed for this dichotomy: 1) either small planets

Figure 1: Bimodal distribution of the exoplanets coming from the Kepler catalogue.

(i.e. $R < 1.8R_{\oplus}$) are unable to form thick atmospheres or 2) they are unable to retain them due to the intense XUV emissions from the young parent stars (Lehmer et al. 2017). The second explanation based on the photoevaporation model seems more plausible as thick atmospheres can be formed by degassing of volatile elements during the early stages of rocky planet (e.g. magma ocean stage).

However, it should be noted that these results and their interpretations are so far only valid for planets with relatively short orbital periods (< 100 days, as shown in Figure 1) as the statistics for planets with longer orbital periods is still insufficient to establish a similar distribution. This last point is explicitly mentioned in these two articles as clear limits of these studies: For instance: “*The radius limit for closely orbiting rocky planets appears to be set at $\sim 1.8R_{\oplus}$ by XUV-driven hydrodynamic escape, but to address the limit in rocky planet size for longer period planets, additional studies on rocky planet formation should be conducted.*”, in the conclusion of Lehmer et al. (2017).

Moreover, these exoplanets are also exposed to fairly high stellar light intensities (more than 10 times the intensity received by the Earth, see Figure 2) from Fulton et al. (2017). Actually, the photoevaporation model, the most widely accepted explanation for the bimodality in the radius distribution, suggests that the vast majority of sub-Neptune sized exoplanets found in the Kepler catalogue (and used in statistical studies) might actually be composed of rocky cores with those above $1.8 R_{\oplus}$ having retained their H-He primordial atmosphere. According to this model, these exoplanets are unlikely to have ice layers as they were formed too close to their star for volatiles to condense. Alternatively, other recent studies suggest that these “mini-Neptunes” ($R = 1.8 - 3.8R_{\oplus}$) from the Kepler catalogue could be composed of large amounts of ice with a thick atmosphere made of supercritical water and not H-He. In both cases, this shows that $1.8-3.8R_{\oplus}$ with orbital periods < 100 days and exposed to more than 10 times light intensity than the Earth would not be life-compatible environments.

Therefore, these statistical studies do not bring any constraints on the composition of exoplanets with longer orbital periods and exposed to smaller stellar light intensities. In fact, planets presenting long orbital periods would be less affected by the intense XUV emissions of the host star, which would not preclude the existence of a continuum between small atmosphere-free and large atmosphere-thick exoplanets, with the ratio of ice/rock increasing with the distance to the

Figure 2: Distribution of planetary sizes and occurrences as a function of stellar intensity.

star. This makes us remark that the existence of large ($> 1.8R_{\oplus}$) water-rich planets with thin atmospheres, as drawn in Figure 5 of the previous manuscript (although we agree that the drawn atmosphere is indeed too large for ice Ih to exist), cannot be excluded when considering planets with orbital periods greater than 100 days and receiving smaller stellar flux.

Unterborn et al. (2018) indeed find 324 K at the bottom of Trappist 1-f ice mantle at 25 GPa with a potential adiabatic temperature of 300 K at the top of the liquid water (according to the Method section), considering 50 wt% H₂O and $R = 1R_{\oplus}$. Such temperature in the HP is very low and seems to come from the use of inaccurate equations of state. Unterborn et al. (2018) mention that they used Stixrude and Bukowinski (1990)’s EOS for liquid water (which is an unconventional choice) and implemented custom approximate EOS for ices Ih, VI and VII but do not give any additional information. In fact, for the same surface conditions, using either IAPWS95 or Bollenger et al. (2019) EOS for liquid water (which are reference EOS) as implemented in the module SeaFreeze (<https://github.com/Bjournaux/SeaFreeze>) gives temperatures of 358 K and 363 K respectively at the bottom of the ocean (2.03 and 2.10 GPa respectively). The ice VI layer would be extremely thin or inexistent, and using the most accurate DFT-based EOS for bcc ices (VII, VII’ and X) from French et al. (2015), we obtain a temperature of 700-710 K at 25 GPa at the bottom of the ice VII layer in the case of Trappist-1f. This result is in much closer agreement with ocean planet models from Sotin et al. (2007) based on the experimental EOS of Fei et al. (1993) for ice VII. The adiabatic profiles described above fall within the grey and blue dotted lines plotted in Figure 3 (the full description of this figure comes later in this response).

Moreover, as represented in Unterborn et al. (2018), a thermal boundary layer is expected between the HP ice mantle and the much hotter rocky mantle. Depending on the shape of such thermal boundary layer (TBL), the temperature in the HP ice can increase by several hundreds to thousands of Kelvin. In the case of Trappist 1-f, the potential adiabatic temperature in the rocky mantle was chosen at 1700 K indicating that the combined temperature range covered in the icy and rocky sides would be ~ 1000 K. Even considering an increase of only 200 K on the icy-side TBL would lead to the formation of HP ice at 25 GPa and ~ 900 K, already within the superionic domain.

In summary, all previous considerations illustrate that a large range of thermodynamic condi-

tions can be spanned within the HP ice layers of water-rich exoplanets and that statistics based on the Kepler exoplanet catalogue have clearly identified limitations that avoid a too large generalization of their results to long-period exoplanets (>100 days) that receive less stellar flux due to a lack of statistics.

Now, we still think that our findings apply to water-rich exoplanets that have a HP ice mantle and potentially habitable PT conditions in a (sub-)surface ocean. Although our DFT calculations are done at 1600 K, the applications of our results is not restricted to this exact temperature. Indeed, our study shows that at 1600 K $\sim 2.5\text{wt}\%$ NaCl can be included in bcc ice, which is superionic in the low pressure part of the studied conditions. This numerical result is in agreement with experiments done at much lower pressures and temperatures (Journaux et al. 2017, Ludl et al., PCCP 2017, and A. Ludl PhD thesis). Therefore we can reasonably assume that the solubility of NaCl in bcc ice remains similar at all intermediate conditions, in particular those relevant for water-rich super-Earths, even considering a maximum pressure of ~ 50 GPa at the bottom of their HP ice mantles (corresponding maximum temperatures of $\sim 600\text{-}1100$ K) as it is considered in most recent studies based on the Kepler catalogue whose potential biases have been discussed above. This consideration makes appropriate the discussion on the implications of our results on the transport of electrolyte in water-rich planets with a (sub-)surface ocean presenting “habitable” conditions.

We modified Figure 3 of the manuscript to show the adiabatic profiles in the H_2O layers of planets presenting no (or very thin) atmospheres and a (sub-)surface ocean with “habitable” conditions. Two particular cases are shown: 1) a surface temperature of ~ 273 K which corresponds to the melting point of pure ice Ih (grey dotted line); 2) a surface temperature of ~ 386 K (blue dotted line) which corresponds to a temperature of ~ 393 K at 0.1 GPa in the ocean (combination of the highest temperature and highest pressure at which extremophile organisms thrive on the Earth).

However, we agree that Figure 5 of the previous manuscript should be modified as the drawn atmosphere looks too thick to allow the formation of an ice Ih layer. The proportions of the different bcc ice phases in the HP ice mantle are also not realistic. Thanks to Reviewer #1, Figure 5 now depicts a better defined case that illustrates our discussion: a $1 M_\oplus$ planet ($\sim 1.3R_\oplus$) with 50 wt% H_2O , no or a very thin atmosphere and a surface temperature ranging between 273 K and 386 K (no ice Ih layer and habitable conditions in part of the ocean). In such case, the PT conditions in the ocean are mild, superionic VII' is reached in the HP ice mantle and the radius planet radius is still below the $1.6\text{-}1.8R_\oplus$ which makes such a thick-atmosphere-free ocean planet still plausible according to the statistical studies based on the Kepler catalogue and allows to avoid the debate on the relevance of these statistics for longer period exoplanets that is out of the scope of the present manuscript. Figure 5 has been modified accordingly (see Figure 4 here). We also modified the discussion, including references to Fulton et al. (2017) and Lehmer et al. (2017), and precisising some sentences. The parameters used to show that the inclusion of salt in bcc ices does not impede its convection remain unchanged.

Finally we rewrote parts of the section *Transport of electrolytes in a high-pressure ice mantle* in order to clarify the objective our analytical approach, its limits and the relevance of the results. The numerical applications have been adapted to the $1 M_\oplus$ planet mentioned above which does not affect our conclusions. In addition, we provide more detail about the analytical modeling in the Methods section that should clarify some parts of the reasoning.

Other comments:

Add references to Fulton et al. (2017) and Lehmer et al. (2017) which are relevant to the Super-Earth/mini-Neptune separation.

Figure 3: This depicts the Fig. 3 of the revised manuscript, in which we added the temperature profiles (adiabats) of hypothetical ocean exoplanets without atmosphere and habitable PT conditions in the surface ocean: i) the grey dotted line corresponds to an adiabatic profile anchored at a surface temperature of 273 K (temperature of crystallization of pure ice Ih); ii) the blue dotted line corresponds to an adiabatic profile anchored at a temperature of 393 K at 0.1 GPa (combination of the highest temperature and pressure conditions at which extremophile organisms live). Circled numbers 1, 2, 3 and 4 indicate the adiabat temperature at the bottom of the HP ice mantle for 1, 2, 3, and 4 M_{\oplus} planets (i.e. 1.3, 1.5, 1.7, 1.9 R_{\oplus}) with 50 wt% H_2O and circled number 1' indicates the conditions of the rocky mantle adiabat at the HP ice/rocks interface as in Sotin et al. (2007). The dotted line linking 1 and 1' conditions represents the range of temperatures covered by the thermal boundary layer at the HP ice/rock interface.

We added both references have been added in the manuscript in the section *Transport of electrolytes in a high-pressure ice mantle*.

I'm surprised that the conductivity and diffusion results are not discussed in the main manuscript. These are important and should be present in the main body, and not only in the supplementary.

Although, the electrical conductivity results are shown in Figure S13 and not in the main manuscript, the evolution of proton diffusion as function of volume and salt concentration is already shown in Figure 2.c, and are presented in the section *Properties and phase boundaries of the salty ice* lines 121-127 with value of diffusion and electrical conductivity explicitly given in the text. In addition, the main result that salty ice has larger proton diffusion and electrical conductivity than pure ice is mentioned again line 131 and in the section *Transport of electrolytes in a high-pressure ice mantle*.

We decided to keep the electrical conductivity results in the Supplementary as they present relatively large error bars. The reason behind this, is that such calculation require extremely long simulations to converge more. Due to their faster convergence, the diffusion coefficients results are less noisy, essentially containing similar information, and are thus shown in the main manuscript.

P.7 regarding the value for ice VI used in equation 1. I would encourage the authors to point out that no ice VII viscosity measurements exists to date. Hopefully this will motivate future experimental effort.

Figure 4: Figure 5 of the revised manuscript showing the internal structure of a hypothetical planet with $1M_{\oplus}$, 50 wt% H_2O , and a surface temperature of 273 K (corresponds to the case 1 with a grey line on Figure 3).

This is a very good point. We added the following sentence in the discussion: "To our knowledge no viscosity measurement in ice VII has been reported yet and we highly encourage any experimental effort in this direction".

The last couple of sentences l.192 are incorrect and should be removed: "If anything, the exothermic character of the VI→VII phase change helps convective flow, as has been well documented for the phase transitions in the upper mantle of Earth's mantle. Salt would then be delivered as such to the overlying ocean" As suggested in Journaux et al. (2017), the exsolution of dense NaCl or NaCl hydrates at the VI-VII boundary will make the salt tend to be gravitationally driven downwards. The latent heat generation as an engine for convection claimed by the authors during the ice VII→VI reaction seems difficult to justify. Indeed the VI-VII reaction is mostly isobaric, and result mainly in an volume change. A very minor entropy/enthalpy change is to be expected from this transition, unlikely to provide any significant thermal energy. Indeed, the VII→VI ΔH is only 394 J/mol at 300K if you believe Dunaeva et al. (2010). Furthermore the salty ice VII→ ice VI+NaCl hydrates latent heat budget remains totally unknown (it could be endothermic), so I would just abstain from making any claims here.

The reviewer is apparently not very familiar with the Earth mantle dynamics literature regarding the effects of phase change on mantle convection. The relevance of the exothermic phase change is not so much in the latent heat part but on the volume change and how it is vertically displaced in regions of temperature higher or lower than average. It is related to the exothermic character of the transition through the Clapeyron relationship. In the present situation, an up-welling hot plume will experience a phase change, and the associated volume increase, at a pressure higher than its surrounding, which will provide an extra buoyancy source to the plume. This has been well studied in the opposite context of possible stratification of the Earth mantle re-

sulting from the endothermic phase change from Ringwoodite to Bridgmanite and ferro-perriclas (see eg Olson, P., & Yuen, D. A. (1982). Thermochemical Plumes and Mantle Phase Transitions. Journal of Geophysical Research, 87, 3993–4002.). But the reviewer is right to state that we do not know how salts affect that phase transition and decided to simply remove that sentence that is not critical to the discussion.

We would like to point out however that the scenario described in Journaux et al (2017) in a cartoon form has not been proven by complete dynamical models and is not more supported than our current claims. We could argue, on the opposite, that ex-solution of salts has little effects on the dynamics. Indeed, for the scenario proposed by Journaux et al. to proceed, you need first to separate the salt grains from the ice matrix which is impossible in the solid state since it requires flow to occur at the grain scale. Observations of poly-crystalline xenoliths on Earth with coexisting mineral grains of various densities prove that point. A dynamical effect is still possible but depends on the density difference between the assemblage of ice and salt crystals and that of ice with dissolved salts, with the same bulk composition.

The experimental observations reporting that salts are incompatible with ice VI needs to be more clearly referenced. Add references to Journaux et al. (2017) at the end of the following sentences : - 1.17: "*the presence of an underlying high-pressure ice mantle might prevent the transport of electrolytes due to their insolubility in ice VI.*" - 1.190: "*... ex-solving NaCl to make a polycrystalline assemblage.*"

We added references to Journaux et al. (2017) to the corresponding sentences.

Figure 5: There is a several issues with this figure, as pointed out in my major comment. If you have ice X and ice VII' present, this cannot be a super-Earth, and the surface temperature cannot allow ice Ih to form. I would remove the figure and replace it with one the DFT results from the supplementary material (e.g. Fig. S.14).

We modified Figure 5 as described and shown above. We think that this figure presents interesting concepts that are worth illustrating and that are related to: 1) the spatial distribution of the electrolyte exchange at the HP ice/ocean interface due to the melting/crystallization boundary; 2) to the effect of a thermal boundary layer on the occurrence of both ice VII'' and a basal ocean that might favour the chemical interaction with the silicate mantle.

In addition, due to the confusion regarding the vocabulary used in our manuscript, we changed the term *icy super-Earth* (that was employed here to describe an exoplanet containing large amounts of ice, not ice Ih in particular) by *water-rich exoplanet* (general) or *water-rich super-Earth* (if $1 < R < 1.6 - 1.8R_{\oplus}$).

Reviewer #2

I would like to thank the authors for putting a lot of effort in revising the manuscript. I suggest that the study can be published as it is.

I have only one last remark, based on the comments of the third reviewer - icy bodies for me also mean water-rich planets covered by an ice shell, independent of if any HP ice would form. Since you mean the opposite here, it would be good to either rephrase or explain directly in the abstract/introduction that the ice refers to the bottom of the water layer and not the surface. But even better would be to rephrase the title of the manuscript to not confuse the reader.

We thank Reviewer #2 for his or her work on our manuscript. As mentioned in the last part of our response to Reviewer #1 we changed the term *icy super-Earth* by *water-rich exoplanet* or

water-rich super-Earth (if $R < 1.6 - 1.8R_{\oplus}$ even if as discussed in response to Reviewer #1 this limit in size might be too restrictive) in order to avoid confusion.

Reviewer #3

The manuscript describes the incorporation of salt (NaCl) in high pressure ice structures (VII, VII', VII'' and X) along an isothermal path of 1600 K using state of the art DFT calculations. The results are then incorporated in a convection model of an ice-giant planet to follow the effect of salt on the convection and on the transport of electrolytes through an icy mantle. I am not an expert of DFT calculation but it seems the work presented here (also judging from the comments of other reviewers) is of remarkable quality, very well documented with all aspects described in the supplementary material. The title only describes the solubility in superionic ices, but actually I found that the discussion concerns all ice structures investigated in the MS from ice VII to ice X, although at 1600 K they could be in the superionic state.

First, we are grateful to Reviewer #3 for reviewing our manuscript and for appreciating the quality of our work. Indeed, the title only mentions the solubility of salt in superionic ices and due to the confusion that it raised we decided to change it to: *Solubility of salt in dense high-temperature ice suggests electrolyte permeability in mantles of water-rich exoplanets* that is presumably more descriptive.

Actually the first layer here to uptake salt from the rocky core is ice X, that may or may not be in this superionic state depending on pressure. The latter will depend on P-T conditions.

The title seems restricted to a certain type of ice giant planet while the authors discuss a multi-layer ice mantle that opens a multitude of questions whether the conditions that are chosen for the convection model are the most common one.

The main idea of the discussion section is to evaluate if the transport of electrolytes in a high-pressure ice mantle by convection is possible. Our strategy is to show that even in the worst case scenario, from a convection perspective, the density increase caused by the addition of salt does not impede the thermal convection. These conclusions are strong enough to question the recent assessment that high-pressure ice mantles would act as chemical insulators between a (sub-surface ocean) and a rocky core (Noack et al., Icarus 2016). Moreover, we think that our analytical approach provides interesting ingredients and considerations that allow to have a first order view on the convection inside water-rich exoplanets presenting a high-pressure ice mantle. Moreover, given the uncertainties regarding the nature of super-Earths and sub-Neptunes, we think that a full numerical modelling would be difficult to constrain, and of course, would require a dedicated study.

In the revised version of the manuscript we clarified the objective of the section *Transport of electrolytes in a high-pressure ice mantle* and adapt the numerical applications to an exoplanet whose properties better match the observational constraints from the Kepler catalogue.

How is this establish that the best adiabat is at 1600 K? if the adiabat is slightly lower, how this influence the convection model (for instance if there is only on layer of superionic ice)? How is the thermal profile established here to obtain a convection?

1600 K is not an adiabat, it is simply an isotherm along which the calculations have been done for convenient reasons: it is the isotherm with the densest grid of points in previous papers on the properties of pure H₂O (Hernandez and Caracas, 2016, 2018), giving a first good basis for the present computations.

Although, the different scenarii are somehow discussed, the case study concerns only one particular type of planet. This is where the mantle dynamic modelling is a bit weaker compared to the DFT part, and the latter could stand alone, and I would join reviewer 1 to say that the paper has a strong orientation on the physics of ices and the modelling could be in a separate study with more details and a variety of cases.

As mentioned in response to the other reviewers during the first revision round, we decided to keep and to enrich the discussion on the effect of salt inclusion on the transport of electrolytes in large water-rich planets with a HP ice mantle, considering different scenarii. As mentioned by the reviewers, an analytical convection model cannot describe all complex phenomena that would occur in high-pressure ice mantles. However, we think that it provides some interesting considerations and opens the path towards more complete numerical models. These analytical considerations have been enriched after the previous revision of the manuscript and its limitations are explicitly mentioned. We hope that the considerations presented in our manuscript will motivate studies with more advanced numerical models in the future.

In the calculation they can incorporate quite an amount of salt in the structure because the liquid is quenched to a crystal by sudden increase in pressure in 12 ps. Large amount of salt in ice was also incorporated experimentally by quenching in temperature a salty water then increasing the pressure and temperature to conduct neutron experiment in ice VII. In both cases it seems there is quite a trick to add large amount of salt but I am wondering how this would actually be the case when there is a solid-solid interface at the CMB of such planet?

We think it is a misunderstanding here. The composition of each calculation is indeed fixed as we simulate a closed system. Therefore, we obtain properties of each individual simulated system at different concentrations of salt, even above the solubility limit. This is what we have done to describe the influence of the salt inclusion on the ice properties. However, to determine the solubility of salt in the ice we used thermodynamic modelling based on the thermodynamic properties of all the systems over the entire range of volume-concentration covered. Such approach is based on equilibrium thermodynamics and does not rely on any trick to include the salt in the ice. Likely, Reviewer #3 was misled by the similarities between the simulations done in the first part of the manuscript (crystallization of a salty solution by quench) and the mentioned experimental approach. Our crystallization simulations are only used to determine the inclusion mechanism of salt in bcc ice (substitution of two H₂O molecules by Na⁺ and Cl⁻).

In a water-rich planet, either the salt would be already present in the ice after the accretion or there would be an influx of the electrolytes in the ice by reaction with the core. As mentioned in our answer to Reviewer #3's last question, it is still unclear if there is a reaction between the HP ice and the core. Moreover, depending on the planet characteristics (size, temperature, water content), either superionic ice or fluid H₂O could be in contact with the rocky core. These two cases could favor reactions as proton diffusion is enhanced in high-temperature ice/fluid which likely results in an acidic behavior in H₂O.

What is the driving force to entrain electrolytes upward to the ocean in this case? It seems from the cartoon that the plume cannot be driven by a low density ice channel, because the salty ice is denser than the normal ice. So this is quite opposite to a rocky mantle where plumes are supposed to be hotter and less dense. In figure 5, it is referred as a thermal pile transport, so a thermal effect, but it is a competition between incorporation of salt (changing the density) and ΔT to make the ice lighter. Is there any evaluation of a threshold for both parameters (XNaCl and T) to maintain the convection?

The driving force for plumes is thermal buoyancy. Our simple analytical calculations show 1) that the thermal Rayleigh number is much greater than its critical value for the onset of convection for a vast diversity of planetary objects and 2) that the incorporation of salts from the rocky core is not sufficient to prevent thermal convection. Clearly, more advanced models are desirable but out of the scope of the present paper.

In such salty plumes how much is transferred to the rest of the icy mantle by diffusion and how much can be transferred to the next ice layer? Is there an estimate of the circulation time through the mantle layers?

In the paper, we provide an estimate of the ice flow velocity for our nominal choice of parameter values and the mathematical expressions to compute ones preferred values for different parameter choices. The diffusion coefficient for salt is not well known but for the value we quote as reasonable and the flow velocity we computed, we can state that very little is transferred by diffusion to the surrounding ice. The ratio between (mostly vertical) convective transfer to (mostly horizontal) diffusion is quantified by the compositional Péclet number, $Pe = vd/D \sim 10^{13}$ with our nominal values. This means that salts would be transported vertically to the next layer, lower pressure ice and eventually the ocean, with negligible amount left behind in the ice layer. The transfer time can readily be estimated from our values of flow velocity as $d/v \sim 4000$ yr, again for our nominal values that can easily be modified using our expressions.

Finally, I would point out that the modelling assumes an infinite reservoir of salt at the base of the mantle, but at such P-T conditions the rocky core should not react very much (or at least there is no data about partitioning between ice and rocks in such conditions) and once the elements of interest have been stripped away from a certain layer, how this is replenished? How can the flux of electrolyte keep constant in such model?

This is a difficult question to answer since there is no data on the reaction between ice and silicates at these conditions and it depends on the dynamics and exact physical properties on both sides of the interface. Convection is likely to occur in the silicate core, which would help to replenish the reservoir of elements of interest, possibly helped by magmatism. In the case where melting occurs at the interface, hydrothermal circulation would be an efficient way to strip the silicate core. An interesting possibility, somewhat arm-wavy at this stage, is that, if ice is superionic, it would possibly have an acidic behaviour that would favour the reaction with the silicate core.

Moreover, our considerations also apply in the situation where the ice has a given concentration of salt (e.g. inherited from the accretion) but has not any additional salt influx from the rocky core. Indeed, our results simply show that even at the solubility limit in the HP ice, the presence of salt would not impede the thermally-driven convection. Therefore, the salt would be transported upward and released to the ocean by melting of the ice. Then, depending on the saturation level of the ocean, either all the salt could be progressively accumulated in the ocean or some of it could precipitate and be entrained again in the convecting HP ice mantle. As mentioned by Reviewer #3, the evolution of the salt concentration in each layer depends in such scenarios depends on the partitioning coefficients between the different envelopes. Determining these partitioning coefficients would be a natural extension of our work.

Overall this a strong and interesting contribution worthy of publication in Nature Communication.

We thank again Reviewer #3 for his or her time and constructive questions.

**REVIEWER COMMENTS**

**Reviewer #1 (Remarks to the Author):**

**I thank the authors for the review of their manuscript. Their detailed and well researched answer is**
**very valuable and has raised some interesting points about the current exoplanet catalogue. The edits**
**are increasing the planetary science quality of the paper significantly. I am still a bit troubled by some**
**of their claims that I think needs to be addressed before the paper is finally published.**

**About the range of applications for exoplanets with orbital periods >100days, the author made a set of**
**very nice points in the response to the reviewers that are unfortunately absent in their edit of the**
**current version of the revised manuscript. This is unfortunate as this could be very beneficial for most**
**readers. The new paragraph between line 181 and 191 is quite confusing and a bit too vague when**
**compared to the reviewer's response. It needs to be rewritten for clarity. I would recommend to detail**
**here the adiabatic calculations (see comments hereafter) and the limits of the present study (see**
**hereafter as well).**

**The main issue that remains is about the details of the calculation for the adiabatic profile are not**
**provided anywhere in the manuscript or the supplementary (not in the version I have been sent). The**
**authors mention "Other scenarios can be tested by plugging in different values in the numerical**
**applications", but nowhere are provided the details to reproduce those calculations. What is the**
**density and size of the silicate/metal core they used for example? Is gravity constant in the**
**hydrosphere or solved for? The authors need to include in the main manuscript the details provided in**
**the following paragraph of their response to reviewers:**

**"In fact, for the same surface conditions, using either IAPWS95 or Bollenger et al. (2019) EOS for**
**liquid water (which are reference EOS) as implemented in the module SeaFreeze**
**(<https://github.com/Bjournaux/SeaFreeze>) gives temperatures of 358 K and 363 K respectively at the**
**bottom of the ocean (2.03 and 2.10 GPa respectively). The ice VI layer would be extremely thin or**
**inexistent, and using the most accurate DFT-based EOS for bcc ices (VII, VII' and X) from French et al.**
**(2015), we obtain a temperature of 700-710 K at 25 GPa at the bottom of the ice VII layer in the case**
**of Trappist-1f"**

**I am also surprised that the adiabatic profiles for 1, 2, 3, and 4 M \oplus planets fall on the same line. As the**
**gravity profile in the hydrosphere are likely to be different for different rocky/metal cores sizes, the**
**adiabatic profiles should depart from each other for different mass. Maybe the effect is small, but**
**unfortunately as no details on the calculations are provided there is no way to reproduce the**
**calculations.**

**Assuming the profiles provided are correct, no details are provided on how the thermal boundary layer**
**is calculated. No supporting literature or calculation are provided to explain or justify the slope and**
**amplitude of the 300-600K jump to 1600K. Maybe it is possible locally, but without any supporting**
**information it is hard to see it more than pure speculation at this point, only to reach the required**
**temperature of the manuscript calculations in a planet with a potentially habitable surface ocean.**

**Even if true (and it needs to be supported or clearly stated that it is an assumption), it is clear that the**
**present results (>1500K) can only apply in the feet of the adiabatic profile for cold enough water-rich**
**exoplanets. Most of the mantle of a water-rich super Earth with a liquid ocean at the surface (and**
**possibly compatible with life as we know it) will be at much lower temperatures. This needs to be**
**clearly stated in the paper. As the author mention in the answer to the reviewers, there are**
**experimental data that show that salts can be incorporated at lower temperatures. So a couple of**
**sentences clearly explaining the limits of the conclusions for the state of the ice mantle would be**
**important here. For example the following sentence present in the reviewer response is very good at**
**explaining the limitations and should be in some form the main manuscript :**

**"Although our DFT calculations are done at 1600 K, the applications of our results is not restricted to**
**this exact temperature. Indeed, our study shows that at 1600 K ~2.5wt% NaCl can be included in bcc**
**ice, which is superionic in the low pressure part of the studied conditions. This numerical result is in**
**agreement with experiments done at much lower pressures and temperatures (Journaux et al. 2017,**
**Ludl et al., PCCP 2017, and A. Ludl PhD thesis). Therefore we can reasonably assume that the solubility**
**of NaCl in bcc ice remains similar at all intermediate conditions, in particular those relevant for water-**
**rich super-Earths, even considering a maximum pressure of ~50 GPa at the bottom of their HP ice**

mantles (corresponding maximum temperatures of ~600-1100 K) as it is considered in most recent
studies based on the Kepler catalogue whose potential biases have been discussed above."

The choice of the 386K surface temperature, above the boiling point at 1 bar is surprising. First, it is
beyond the definition of the, disputable but widely referenced, "habitable zone". No extremophile
"thrive" at this temperature as it represent the upper limit for bacterial growth (395K, Kashefi and
Lovley, 2003; Takai et al., 2008). Furthermore, at such temperature, it is likely a thick water vapor
atmosphere is present increasing the pressure (and temperature by greenhouse effect) quite
significantly. I don't think we can demonstrate that there is any planet without a thick atmosphere
with such conditions, even less a planet interesting for habitability. If the authors want to keep an
upper limit here, I would recommend to use a lower upper temperature like 350K which is often used
in the Exoplanet literature.

I would also suggest to rephrase the abstract to temper a bit the astrobiology implication.

**Small comments:**

The edits made to figure 3 are unfortunately confusing as the description is missing in the the caption
about the adiabatic profiles in the manuscript (it is in the reviewer response though).

**L.187-188:** the sentence "The present case is interesting as it results in life-compatible thermodynamic
conditions in the surface ocean", is a bit misleading and could be controversial in astrobiology. A
sentence like :

"The present case is interesting as it results in a surface ocean with reasonable conditions for
potentially sustaining life as we know it" would be preferable. The authors could also cite here Cockell,
C. s. et al. Habitability: A Review. *Astrobiology* 16, 89–117 (2016).

Once this is address I think this very good manuscript will be ready for publication. I congratulate the
authors for this amazing work and I hope they will forgive the slight severity of some of my comments.
I really want this amazing work to be taken seriously by the planetary science and astrobiology
community.

Point-by-point response to the reviewers

We thank the Reviewer #1 for its positive feedback on the last revisions and discussions. In the following, we address his remaining concerns and suggestions. Overall, this communication has largely improved the quality of the manuscript regarding planetary implications.

About the range of applications for exoplanets with orbital periods >100 days, the author made a set of very nice points in the response to the reviewers that are unfortunately absent in their edit of the current version of the revised manuscript. This is unfortunate as this could be very beneficial for most readers. The new paragraph between line 181 and 191 is quite confusing and a bit too vague when compared to the reviewer's response. It needs to be rewritten for clarity. I would recommend to detail here the adiabatic calculations (see comments hereafter) and the limits of the present study (see hereafter as well).

As suggested by the reviewer, we re-organized and reformulated the paragraph between lines 181 and 191 of the previous manuscript (see text colored in red in the new version of the manuscript).

A discussion about the limitations of the statistical studies showing the exoplanet radius valley has been added in Additional Information and is referenced in the above paragraph. The reason for not including this interesting discussion in the main manuscript is to keep it focused on how the salt would be transported or affect the ice mantle convection.

Details about the calculation of the adiabatic profiles have also been added in Figure 3 caption and Additional Information. See following answers.

The main issue that remains is about the details of the calculation for the adiabatic profile are not provided anywhere in the manuscript or the supplementary (not in the version I have been sent). The authors mention "Other scenarios can be tested by plugging in different values in the numerical applications", but nowhere are provided the details to reproduce those calculations. What is the density and size of the silicate/metal core they used for example? Is gravity constant in the hydrosphere or solved for? The authors need to include in the main manuscript the details provided in the following paragraph of their response to reviewers: "In fact, for the same surface conditions, using either IAPWS95 or Bollerger et al. (2019) EOS for liquid water (which are reference EOS) as implemented in the module SeaFreeze (<https://github.com/Bjournalux/SeaFreeze>) gives temperatures of 358 K and 363 K respectively at the bottom of the ocean (2.03 and 2.10 GPa respectively). The ice VI layer would be extremely thin or inexistent, and using the most accurate DFT-based EOS for bcc ices (VII, VII' and X) from French et al. (2015), we obtain a temperature of 700-710 K at 25 GPa at the bottom of the ice VII layer in the case of Trappist-1f". I am also surprised that the adiabatic profiles for 1, 2, 3, and 4 M_{\oplus} planets fall on the same line. As the gravity profile in the hydrosphere are likely to be different for different rocky/metal cores sizes, the adiabatic profiles should depart from each other for different mass. Maybe the effect is small, but unfortunately as no details on the calculations are provided there is no way to reproduce the calculations.

First, the sentence "Other scenarios can be tested by plugging in different values in the numerical applications" line 186 has been mistakenly placed in the wrong paragraph during the last revision. This sentence actually applies to the evaluation of the thermal and compositional

convection (from line 192 of the previous version). The numerical values (core radius R_c , heat flux from the core q_c , etc) chosen in this last analysis and given in equations (1) (calculation of the Rayleigh number) and (2) (calculation of the buoyancy number) can indeed be changed and adapted to different planetary cases if needed. In the present manuscript, the chosen numerical values correspond to the investigated theoretical planets whose adiabatic profiles are given above. All the details to reproduce this last analysis were already given in the paragraph between lines 192 and 226 of the previous manuscript and in "Geodynamical scaling" method subsection. Finally, the sentence "*In the present case, geometric factors of the planet are from Sotin et al. [46], but these values can be changed depending on the planet of application.*" already refer to this idea and for this reason we removed the sentence "*Other scenarios...*" from the manuscript to avoid confusion.

Regarding the adiabatic profiles presented in Figure 3, we agree that not enough details were given in the previous version of the manuscript about the calculations and limitations. Moreover, the grey profile which was supposed to represent our calculated adiabat for a surface temperature of 273 K was actually the P-T profile calculated by Sotin et al. (2007) for a surface temperature of 300 K and based on an extrapolated and modified equation of state from Fei et al. (1993). The blue profile corresponds to the description given in the previous answer to the reviewer and is the adiabat for a surface temperature of 386 K. We apologize for this erroneous figure. This has been corrected in the new version of Figure 3, in which the corresponding profiles are calculated based on the equation of state of French et al. (2015) in the bcc ice domain. Three profiles are now shown, one is calculated for a surface temperature of 273 K, another for a surface temperature of 350 K considering the reviewer's recommendation, and a third one at 300 K to match the conditions imposed in the model from Sotin et al. (2007) and which we use in the discussion.

About the calculation itself, the adiabatic profiles are only computed in the H₂O layer by using the equations of state of H₂O phases to determine the pressure-temperature conditions. In the liquid layer, the adiabatic profile is computed by finding the PT conditions whose entropy equals the entropy at the surface of the planet in the three different cases shown in Figure 3 (surface temperatures of 273 K, 300 K and 350 K). The intersection of the liquid layer adiabatic profile with the melting line of either ice VI (colder case) or ice VII serves as anchor point for the adiabats of these phases. Similarly in the colder case, the intersection of ice VI adiabat with the ice VI/ice VII boundary serves as anchor point for the adiabatic profile in the ice VII layer. In the H₂O layer, the adiabats of planets with different sizes presenting the same surface conditions should share PT profiles in the overlapping pressure ranges. However, the pressure evolution as function of depth and mass should indeed be different due to the effect of gravity. Here, we did not compute planetary models that solve consistently the thicknesses of the different envelopes and the mass, gravity, density, pressure and temperature as done in Sotin et al. (2007) for instance. Instead, we placed both the P-T conditions at the base of the ice layer (labels 1, 2, 3, 4 in Figure 3 of the previous version) and the conditions in the rocky underlying envelope (label 1' and extent of the possible thermal boundary layers in Figure 3 of the previous version) based on the models of Sotin et al. (2007) considering ocean exoplanets with Earth-like cores and using an extrapolated EOS for bcc ice (Fei et al. 1993). Consequently, thermal boundary layers were not computed either and the temperature jump between the base of the ice layer and the top of the rocky mantle comes from Sotin et al. (2007).

In the new Figure 3, we kept the labels 1, 2, 3, 4 for indicating the approximate pressures at the bottom of the HP ice layer based on those found in Sotin et al. (2007) considering water-rich planets with Earth-like cores. Although Sotin et al. (2007) did not use the same EOS for bcc ice, those pressures should change by no more than $\sim 10\%$ and we think that those labels represent a useful when discussing the implications of our work, illustrated in the case of a 50 wt% H₂O-rich

exoplanet. The following lines have been added to the caption:

”Blue dashed lines show adiabatic profiles in the H_2O layer of hypothetical ocean exoplanets with habitable PT conditions in the surface ocean and no or thin atmosphere (see Additional Information). The different profiles are anchored at surface temperatures of 273 K (temperature of crystallization of pure ice Ih), 300 K and 350 K (representative temperature for an optimal growth of thermophile and hyperthermophile organisms [36]) respectively. Circled numbers 1, 2, 3 and 4 indicate approximate pressures at the bottom of the HP ice mantle for 1, 2, 3, and 4 M_{\oplus} planets (i.e. $\sim 1.3, 1.5, 1.7, 1.9 R_{\oplus}$) with 50 wt% H_2O and Earth-like cores as calculated in Sotin et al. (2007) for a surface temperature of 300 K.”

In Additional Information we added: *The adiabatic profiles shown in Figure 3 are only computed in the H_2O layer by using the equations of state of H_2O phases to determine the pressure-temperature conditions (ref. [22] or [23] for liquid water, ref. [24] for ice VI and ref. [5] for bcc ices). Due to the limited range of temperature accessible with the reference EOS from Bollengier et al. (2019) [22], we used the EOS from Brown et al. (2018) [23] to compute the thermodynamic states encountered in liquid water at the conditions when considering a surface temperature of 350 K. Both EOS were used as implemented in the Python module SeaFreeze (<https://github.com/Bjournaux/SeaFreeze>). In the liquid layer, the adiabatic profile is computed by finding the PT conditions whose entropy equals the entropy at the surface of the planet in the three different cases shown in Figure 3 (surface temperatures of 273 K, 300 K and of 350 K). The intersection of the liquid layer adiabatic profile with the melting line of either ice VI (colder case) or ice VII serves as anchor point for the adiabats of these phases. Similarly in the colder case, the intersection of ice VI adiabat with the ice VI/ice VII boundary serves as anchor point for the adiabatic profile in the ice VII layer. Table 1 gives the PT conditions and entropy value that serves as anchor points for the different parts of the adiabatic profiles. The P-T conditions at the base of the ice layer (labels 1, 2, 3, 4 in Figure 3) were positioned based on the planetary models computed in Sotin et al. (2007) representing ocean planets with 50 wt% H_2O , Earth-like cores and 1 to 4 Earth masses. Although Sotin et al. (2007) used a different equation of state for bcc ices in their planetary models (extrapolation and modification of Fei et al. 1993), the pressure at the bottom of the ice layer should not change by more than $\sim 10\%$ when using the equation of state from French et al. (2015).*

Table 1: *Anchor points for the isentropes in the different H_2O layers for the profiles shown in Figure 3.*

Boundary	T (K)	p (GPa)	s ($J.kg^{-1}.K^{-1}$)	Phase	EOS
Surface	273	0	0.104	Liquid	Bollengier et al. (2019)
Liquid/VI	298	0.97	-1063.219	VI	Journaux et al. (2020)
VI/VII	322	2.20	2498.743	VII	French et al. (2015)
Surface	350	0	1038.044	Liquid	Brown et al. (2018)
Liquid/VII	474	3.90	3413.730	VII	French et al. (2015)

Assuming the profiles provided are correct, no details are provided on how the thermal boundary layer is calculated. No supporting literature or calculation are provided to explain or justify the slope and amplitude of the 300-600K jump to 1600K. Maybe it is possible locally, but without any supporting information it is hard to see it more than pure speculation at this point, only to reach the required temperature of the manuscript calculations in a planet with a potentially habitable surface ocean.

We disagree with this statement. In previous Figure 3, the temperature jump between the base of the ice layer and the top of the rocky mantle was not a trick to match habitable conditions at the surface, and this for the following reasons:

1. The isentropic profile computed in our study in the H₂O layer was constructed based on the entropy of liquid water at the surface conditions, and therefore does not depend on the conditions in the rocky mantle.
2. Based on the excellent agreement between the PT profiles of the H₂O layer computed in our study and in Sotin et al. 2007, we decided to use the pressure-mass relations given in Figure 4 of Sotin et al. 2007 (computed considering ocean planets with Earth-like cores and surface temperatures of 300 K) to set the pressure at the bottom of the ice layer (labels 1, 2, 3, 4 in Figure 3) and at the top of the rocky mantle (only label 1' visible) for each planet. The resulting temperature jump between the two layers was identical to the one used in Sotin et al. 2007 which is based on the Earth's case (800 K at the Earth's core-mantle boundary based on Poirier et al. 1994).

However, we agree that the amplitude of the temperature jump at the ice-silicate boundary would certainly influence the size and the composition of the thermal boundary layer. Regarding the slope of the thermal boundary layer, it was drawn as not infinite simply to highlight the presence of a thermal boundary layer, whatever its thickness.

We recognize that due to a missing part in Figure 3 previous version of the manuscript the origin and signification of these profiles were not clear and confusing, and we apologize for this. The corrected version should provide all the necessary information. In addition, we added in the caption to Figure 5 the following sentence to clarify that the presence of both ice VII'' and fluid regions at the bottom of the ice mantle depends on the temperature profile in the thermal boundary layer which is not investigated in the present study: "*Label C illustrates a possible thermal boundary layer at the bottom of the icy mantle at the contact with the hotter rocky mantle.*" Moreover, we modified Figure 5 by replacing "*In the thermal boundary layer*" by "*Possible thermal boundary layer*" in item C.

Even if true (and it needs to be supported or clearly stated that it is an assumption), it is clear that the present results (>1500K) can only apply in the feet of the adiabatic profile for cold enough water-rich exoplanets. Most of the mantle of a water-rich super Earth with a liquid ocean at the surface (and possibly compatible with life as we know it) will be at much lower temperatures. This needs to be clearly stated in the paper. As the author mention in the answer to the reviewers, there are experimental data that show that salts can be incorporated at lower temperatures. So a couple of sentences clearly explaining the limits of the conclusions for the state of the ice mantle would be important here. For example the following sentence present in the reviewer response is very good at explaining the limitations and should be in some form the main manuscript : "Although our DFT calculations are done at 1600 K, the applications of our results is not restricted to this exact temperature. Indeed, our study shows that at 1600 K ~2.5wt% NaCl can be included in bcc ice, which is superionic in the low pressure part of the studied conditions. This numerical result is in agreement with experiments done at much lower pressures and temperatures (Journaux et al. 2017, Ludl et al., PCCP 2017, and A. Ludl PhD thesis). Therefore we can reasonably assume that the solubility of NaCl in bcc ice remains similar at all intermediate conditions, in particular those relevant for water-rich super-Earths, even considering a maximum pressure of ~50 GPa at the bottom of their HP ice mantles (corresponding maximum temperatures of ~600-1100 K) as it is considered in most recent studies based on the Kepler catalogue whose potential biases have been discussed above."

We modified the following sentence that was, to our opinion, already emphasizing the applications of our results were not restricted to 1600 K:

"The combination of our high-pressure and high-temperature results with previous studies done at milder conditions [8, 13] shows that bcc ices are able to include non-negligible amounts of salt in their structure over a broad range of conditions relevant for the interiors of water-rich super-Earths [36] ($1R_{\oplus} < R < 1.8R_{\oplus}$), mini-Neptunes ($1.8R_{\oplus} < R < 4R_{\oplus}$), and Neptune-like exoplanets [39, 40] according to the classification made in ref. [37, 38] based on the Kepler exoplanet catalogue."

into:

"Although our DFT calculations are done at 1600 K, the applications of our results is not restricted to this exact temperature. Indeed, the combination of our high-pressure and high-temperature results with previous studies done at milder conditions [8, 13] shows that bcc ices are able to include non-negligible amounts of salt in their structure over a broad range of conditions relevant for the interiors of water-rich super-Earths [36] ($1R_{\oplus} < R < 1.8R_{\oplus}$ according to ref. [37, 38]), mini-Neptunes ($1.8R_{\oplus} < R < 4R_{\oplus}$), and Neptune-like exoplanets [39, 40]."

The choice of the 386K surface temperature, above the boiling point at 1 bar is surprising. First, it is beyond the definition of the, disputable but widely referenced, "habitable zone". No extremophile "thrive" at this temperature as it represent the upper limit for bacterial growth (395K, Kashefi and Lovley, 2003; Takai et al., 2008). Furthermore, at such temperature, it is likely a thick water vapor atmosphere is present increasing the pressure (and temperature by greenhouse effect) quite significantly. I don't think we can demonstrate that there is any planet without a thick atmosphere with such conditions, even less a planet interesting for habitability. If the authors want to keep an upper limit here, I would recommend to use a lower upper temperature like 350K which is often used in the Exoplanet literature.

We agree with the reviewer's suggestion and computed new profiles considering a surface temperature of 350 K. See also our answer to the first and second comments. The corresponding profile on Figure 3 has been modified accordingly.

I would also suggest to rephrase the abstract to temper a bit the astrobiology implication.

We modified the first sentence of the abstract:

As most biological reactions occur in aqueous solutions, habitability of water-worlds is related to the presence of a (sub-)surface ocean, enriched in nutrients and/or electrolytes. However, in water-rich exoplanets, the presence of a high-pressure ice mantle may hinder the exchange and transport of electrolytes between various liquid and solid deep layers.

by:

Electrolytes play an important role on the internal structure and dynamics of water-rich satellites and potentially water-rich exoplanets. However, in the latter, the presence of a large high-pressure ice mantle is thought to hinder the exchange and transport of electrolytes between various liquid and solid deep layers. [...]

The edits made to figure 3 are unfortunately confusing as the description is missing in the the caption about the adiabatic profiles in the manuscript (it is in the reviewer response though).

We thank the reviewer for noting this oversight. We added the description of the adiabatic profiles in Fig. 3 caption.

L.187-188: the sentence "The present case is interesting as it results in life-compatible thermodynamic conditions in the surface ocean", is a bit misleading and could be controversial in astrobiology. A sentence like : "The present case is interesting as it results in a surface ocean with reasonable conditions for potentially sustaining life as we know it" would be preferable. The authors could also cite here Cockell, C. s. et al. Habitability: A Review. *Astrobiology* 16, 89–117 (2016).

We followed the reviewer' suggestion and changed the manuscript accordingly.

Once this is address I think this very good manuscript will be ready for publication. I congratulate the authors for this amazing work and I hope they will forgive the slight severity of some of my comments. I really want this amazing work to be taken seriously by the planetary science and astrobiology community.

We thank the reviewer for these encouraging comments and appreciate all his/hers efforts in improving our manuscript.

**REVIEWERS' COMMENTS**

Reviewer #1 (Remarks to the Author):

I would like to sincerely thanks the authors for this thorough review of the manuscript and their honest and
dedicated efforts to answer all of my comments.

I am now convinced by all their responses and edits. I think this very good manuscript is ready for publication as
is.

I am also convinced that this is a very important contribution and I would support Nature communications doing an
editorial about it.

Answer to the reviewer

Reviewer #1:

I would like to sincerely thank the authors for this thorough review of the manuscript and their honest and dedicated efforts to answer all of my comments.

I am now convinced by all their responses and edits. I think this very good manuscript is ready for publication as is.

I am also convinced that this is a very important contribution and I would support Nature communications doing an editorial about it.

We thank Reviewer #1 for these encouraging comments and his/her efforts in improving this manuscript.